# Boosting Out-of-Distribution Detection with Multiple Pre-trained Models

## Abstract

Out-of-Distribution (OOD) detection, i.e., identifying whether an input is sampled from a novel distribution other than the training distribution, is a critical task for safely deploying machine learning systems in the open world. Recently, post hoc detection utilizing pre-trained models has shown promising performance and can be scaled to large-scale problems. This advance raises a natural question: Can we leverage the diversity of multiple pre-trained models to improve the performance of post hoc detection methods? In this work, we propose a detection enhancement method by ensembling multiple detection decisions derived from a zoo of pre-trained models. Our approach uses the p-value instead of the commonly used hard threshold and leverages a fundamental framework of multiple hypothesis testing to control the true positive rate of In-Distribution (ID) data. We focus on the usage of model zoos and provide systematic empirical comparisons with current state-of-the-art methods on various OOD detection benchmarks. The proposed ensemble scheme shows consistent improvement compared to single-model detectors and significantly outperforms the current competitive methods. Our method substantially improves the relative performance by $65.40\%$ and $26.96\%$ on the CIFAR10 and ImageNet benchmarks.

## 1 Introduction

Deep neural networks have achieved empirical success in many applications, but generalization robustness has always been a thorny problem in deep learning. A sophisticated and well-trained deep neural network can provide excellent test performance on identically distributed (ID) test data but may fail to make accurate predictions on inputs from outside the training distribution Nguyen et al. (2015). This poses a big obstacle to the generalization of deep neural network models. Especially in safety-critical applications, it is better to identify out-of-distribution (OOD) inputs ahead of time rather than letting the model make predictions that may be unreliable.

On the basis of pre-trained deep neural networks, many recent works on post hoc OOD detection have proposed diverse score functions to distinguish OOD samples utilizing the output probability, logits, gradients, and features of the pre-trained classifier. At the same time, some works also propose new training strategies to encourage the network to learn more features that may not be relevant to the OOD classification task. For example, MSP (Hendrycks & Gimpel, 2017) uses the maximum softmax probability, Energy score (Liu et al., 2020) considers the logits, and GradNorm (Huang et al., 2021) employs the vector norm of gradients. Based on these frameworks, several improved methods such as ODIN (Liang et al., 2018), Adjusted Energy Score (Lin et al., 2021), ReAct (Sun et al., 2021) are proposed to enhance the performance of OOD detection. These score functions above measure the similarity between a test input and the training (ID) data through pre-trained feature extractors or classifiers. There are also many distance-based algorithms that directly quantify the distance of samples in the embedding space extracted from a pre-trained model and regard a test input as an OOD sample when it is far from the ID data. Lee et al. (2018) assumes the conditional distribution of features given the class label is a Gaussian distribution and derives a confidence score based on the Mahalanobis distance. SSD (Sehwag et al., 2020) considers self-supervised pre-training and a Mahalanobis distance. Tack et al. (2020) uses contrastive learning with distributionally-shifted augmentations for pre-training and proposes a detection score specific to their training scheme. Sun et al. (2022) studies the nearest-neighbor distance and demonstrates the efficacy of non-parametric modeling of the feature distribution for OOD detection tasks.

The performance of post hoc detection highly depends on the quality of pre-training. The most commonly used model architectures in OOD detection include convolutional networks such as ResNet (He et al., 2016), DenseNet (Huang et al., 2017) and Wide-ResNet (Zagoruyko & Komodakis, 2016), and of course Transformer models such as Swin (Liu et al., 2022) or ViT (Dosovitskiy et al., 2021). In general, the pre-trained models focus on the features related to classification tasks and the learnt representation may be insufficiently rich for OOD detection. Therefore, researchers have proposed ideas such as contrastive learning (Winkens et al., 2020; Tack et al., 2020), adversarial training Biggio & Roli (2018); Miller et al. (2020); Chalapathy & Chawla (2019) , outlier exposure (Hendrycks et al., 2018; Papadopoulos et al., 2021) or other auxiliary artificially synthesized data (Lee et al., 2017) and auxiliary loss function (Vyas et al., 2018) to encourage models to learn high-level, task-agnostic and comprehensive features, which makes the model more robust and efficient in the downstream detection task. These models trained with different architectures and training strategies can extract diverse features that may complement each other well. So, a natural question is raised:

*Can we leverage the diversity of multiple pre-trained models to improve*
*the performance of post hoc OOD detectors?*

To answer this question, we first build a model zoo that captures as many properties of the input as possible and remains sensitive to distributional changes. Then we reformulate the OOD detection task to check whether there exists a model in the model zoo that can identify the test input as an OOD sample. Section 3.1 shows that the naive ensemble of multiple OOD detection decisions cannot maintain the true positive rate of the ID data (TPR). Therefore, we propose an ensemble scheme to integrate the results of multiple OOD detectors and provide theoretical guarantees that our method can keep TPR at the target level. In Section 4, we also report the empirical TPR of our method, which is close to the target TPR level.

Ensembling is not new to OOD detection. Morningstar et al. (2021) combines multiple test statistics from generative models to differentiate ID and OOD data. Haroush et al. (2022) uses both the Simes' method and Fisher's method to summarize p-values computed for each channel and layer of a deep neural network. Bergamin et al. (2022) shows that combining different types of test statistics using Fisher's method overall leads to a more accurate out-of-distribution test. Recently, Magesh et al. (2022) proposes an ensemble framework that combines any number of different test statistics using the Benjamini–Yekutieli procedure (Benjamini & Yekutieli, 2001) and a conformal p-value estimator (Vovk et al., 1999). In this work, we develop a simple and fundamental ensemble scheme for using model zoos in OOD detection and name our method **Z**oo-based **O**OD **D**etection **E**nhancement (**ZODE**). Our method directly estimates the p-values according to its definition and employs the Benjamini–Hochberg procedure (Benjamini & Hochberg, 1995) to control TPR. Then, we provide theoretical guarantees and empirical validation to show that ZODE can maintain the TPR close to its target level. On the other hand, we focus on the settings of the model zoo and conduct systematic experiments to demonstrate the superiority of our approach. First, we show that ZODE can consistently improve current OOD detectors. Second, by comparing single-model detectors with the ZODE-ensembled detector, we find that ZODE can exploit the diversity of multiple pre-trained models and leverage complementarity among single-model detectors. Finally, our approach significantly improves current SOTA performance.

We summarize our contributions as follows:

- We provide novel insights into OOD detection from the perspective of the model zoo. We propose an enhancement scheme, ZODE, for OOD detection by exploiting the diversity of pre-trained models. The proposed method is inspired by a simple and fundamental framework of multiple hypothesis testing. Our theoretical results and experiments clearly show that ZODE can leverage the complementarity among single-model detectors to improve performance.

- We point out that the naive ensemble of multiple OOD detectors leads to lower TPR. Then we provide theoretical analysis and empirical validation to demonstrate that our proposed method can maintain TPR well under the settings of the model zoo.

- Extensive experiments show that our method can effectively and consistently improve the power of identifying OOD samples. On a commonly used CIFAR10 benchmark, our method significantly improves the SOTA result of the average false positive rate from 11.07% to 3.83%. For a challenging OOD detection task based on ImageNet, we show

that our method is scalable to large-scale problems and significantly improves the SOTA result of the average false positive rate from $38.47\%$ to $28.10\%$.

## 2 PRELIMINARIES

Out-of-Distribution Detection aims to check whether a test input is generated from the training distribution or not. It is a one-sample hypothesis testing problem if we can only access the training data. We denote $\mathcal{X}$ and $\mathcal{Y}$ as the input and label space respectively and let $\mathcal{P}_{id}$ be the training distribution over $\mathcal{X} \times \mathcal{Y}$. Suppose that $\phi(\mathbf{x})$ is a neural network trained on data drawn from $\mathcal{P}_{id}$ to predict the label of input $\mathbf{x} \in \mathcal{X}$. Let $\mathcal{D}_{id}$ denote the marginal distribution on $\mathcal{X}$. Then we call $\mathbf{x} \sim \mathcal{D}_{id}$ an in-distribution (ID) sample, otherwise, we identify it as an "unknown" input, called out-of-distribution (OOD) data. At test time, OOD detection distinguishes OOD samples and ID samples by using a decision function:

$$G(\mathbf{x}^*) = \begin{cases} ID & S(\mathbf{x}^*) \geq \lambda; \\ OOD & S(\mathbf{x}^*) < \lambda; \end{cases} \tag{1}$$

where $\mathbf{x}^*$ is a test input, $S(\cdot)$ is a score function that gives higher scores for ID data and lower for OOD data, and $\lambda$ is the threshold. In this work, we consider post hoc OOD detection in which the score function $S$ is derived from a pre-trained classifier $\phi$, i.e. $S(\mathbf{x}^*) = S(\mathbf{x}^*; \phi)$.

We denote $F(s; \phi)$ as the distribution of $S(\mathbf{x}; \phi)$ with $\mathbf{x} \sim \mathcal{D}_{id}$ and any pre-trained model $\phi$. Then, if $\mathbf{x}^*$ is an ID sample, the score $S(\mathbf{x}^*; \phi)$ is an ID value following the distribution $F(s; \phi)$. Therefore, given a pre-trained model zoo $\mathcal{M} = \{\phi_1, \ldots, \phi_m\}$, we strengthen the OOD detection problem to:

*Is there $\phi \in \mathcal{M}$ that would allow us to identify $\mathbf{x}^*$ as an OOD sample?*

In this work, we proposed an approach to achieve the goal of this OOD detection problem.

## 3 METHODOLOGY

### 3.1 NAIVE ENSEMBLE CANNOT MAINTAIN TPR

To leverage the model zoo $\mathcal{M}$ for OOD detection, a straightforward way is to execute the detection procedure in Eq.(1) based on each pre-trained model:

$$G(\mathbf{x}^*; \phi) = \begin{cases} ID & S(\mathbf{x}^*; \phi) \geq \lambda_\phi; \\ OOD & S(\mathbf{x}^*; \phi) < \lambda_\phi; \end{cases}$$

and identify $\mathbf{x}^*$ as an OOD sample if there exists $\phi \in \mathcal{M}$ such that $G(\mathbf{x}^*; \phi) = OOD$, i.e.,

$$G(\mathbf{x}^*; \mathcal{M}) = \begin{cases} ID & \text{if } S(\mathbf{x}^*; \phi) \geq \lambda_\phi, \ \forall \phi \in \mathcal{M}; \\ OOD & \text{if } S(\mathbf{x}^*; \phi) < \lambda_\phi, \ \exists \phi \in \mathcal{M}; \end{cases} \tag{2}$$

In other words, $\mathbf{x}^*$ is classified as an ID sample only if all detectors $G(\mathbf{x}^*; \phi_i)$, $\phi_i \in \mathcal{M}$ agree that $\mathbf{x}^*$ is an ID sample. However, this simple approach is not easy to control the true positive rate of the ID data (TPR). In practice, the threshold $\lambda_\phi$ is chosen so that a high fraction (e.g. $95\%$) of ID data is correctly identified. We denote the target level of the true positive rate of the ID data as $\text{TPR}_0$ and write $\alpha = 1 - \text{TPR}_0$. Therefore, each detector $G(\mathbf{x}^*; \phi_i)$ has a $\alpha$ probability of misidentifying an ID sample as an OOD sample. When ensembling multiple single-model detectors, the probability of making mistakes also accumulates. It is easy to see that **the detector $G(\mathbf{x}^*; \mathcal{M})$ can misidentify an ID sample as an OOD sample with probability more than $\alpha$, specifically $1 - (1 - \alpha)^m$ when detectors are independent.** As more and more pre-trained models become available, this error probability of $G(\mathbf{x}^*; \mathcal{M})$ increases until it becomes $100\%$. This implies that the naive ensembled detector cannot maintain the target TPR level. On the other hand, by fixing $m$, we can assign a low probability to $\alpha$ to make sure $1 - (1 - \alpha)^m = 5\%$. In this case, $\text{TPR}_0$ should be very large and even close to $1$. This greatly reduces the probability of successfully identifying OOD data, as each single-model detector becomes very conservative and can only identify extreme OOD data. In this work, we develop an ensemble scheme that can maintain the target TPR level while keeping a high probability of successfully identifying OOD data.

## 3.2 USING P-VALUE FOR OOD DETECTION

However, directly integrating score functions is uninterpretable and lacks theoretical guarantees. Therefore, we use the p-value for OOD detection. P-value (Abramovich & Ritov, 2013) is defined in the framework of statistical hypothesis testing. In OOD detection, the p-value is a probability measure that quantifies how extreme the observed score is when the input is an ID sample (Cai & Koutsoukos, 2020; Morningstar et al., 2021; Haroush et al., 2022; Bergamin et al., 2022; Magesh et al., 2022; Kaur et al., 2022). For example, we identify an input $\mathbf{x}$ as an OOD sample (reject the null hypothesis) when the observed detection score $S(\mathbf{x})$ is smaller than a critical value $\gamma$. Given a test sample $\mathbf{x}^*$, the lower value of $S(\mathbf{x}^*)$, the more likely $\mathbf{x}^*$ is not drawn from the training distribution. Hence, the p-value of $\mathbf{x}^*$ is the probability that $S(\mathbf{x})$ is less than $S(\mathbf{x}^*)$ under the ID distribution, that is,

$$\text{P-value of } \mathbf{x}^* = \mathbb{P}\big(S(\mathbf{x}) \leq S(\mathbf{x}^*) \big| \mathbf{x} \sim \mathcal{D}_{id}\big). \tag{3}$$

In general, if the p-value of $\mathbf{x}^*$ is less than $0.05$, we can determine that $\mathbf{x}^*$ is an OOD sample at the significance level $0.05$. In Appendix C, we show that using the p-value is equivalent to using the hard threshold $\lambda$ in Eq. (1).

Suppose the test input $\mathbf{x}^*$ is an ID sample that $\mathbf{x}^* \sim \mathcal{D}_{id}$ and the detection score $S(\mathbf{x}^*)$ is a continuous random variable. We write $p_0$ as the p-value of $\mathbf{x}^*$ and let $F(s)$ be the cumulative distribution function of $S(\mathbf{x})$ with $\mathbf{x} \sim \mathcal{D}_{id}$. Then we have $p_0 = \mathbb{P}\big(S(\mathbf{x}) \leq S(\mathbf{x}^*) \big| \mathbf{x} \sim \mathcal{D}_{id}\big) = F(S(\mathbf{x}^*))$. It follows from the continuity of $S(\mathbf{x}^*)$ and Lemma 21.1 of Van der Vaart (2000) that

$$\begin{aligned}
\mathbb{P}(p_0 < \alpha) &= 1 - \mathbb{P}\big(F(S(\mathbf{x}^*)) \geq \alpha\big) \\
&= 1 - \mathbb{P}\big(S(\mathbf{x}^*) \geq F^{-1}(\alpha)\big) = F(F^{-1}(\alpha)) = \alpha.
\end{aligned}$$

This implies that **the p-value of $\mathbf{x}^*$ follows a uniform distribution** $U[0, 1]$. In the following, we will use this property to develop an ensemble scheme (Theorem 1 and Lemma 2).

## 3.3 TPR CONTROLLING FOR ENSEMBLE

According to Eq. (3), the p-value relies on the score function $S(\mathbf{x})$, which is derived from a pre-trained model $\phi$, i.e. $S(x) = S(x; \phi)$. Given one pre-trained model, we can construct a score function and compute the p-value of a test input. But when multiple pre-trained models are accessible, how to fuse the single-model results to leverage the diversity of multiple pre-trained models while strictly maintaining TPR on ID data?

We borrow the idea of the Benjamini-Hochberg procedure (Benjamini & Hochberg, 1995) and propose an ensemble scheme for OOD detection via p-value correction. Consider a model zoo with $m$ pre-trained models: $\mathcal{M} = \{\phi_1, \phi_2, \ldots, \phi_m\}$ and a score function $S(x; \phi)$. Given a test input $\mathbf{x}^*$ and a pre-trained model $\phi_i$, we compute the score value $S(\mathbf{x}^*; \phi_i)$ and obtain the corresponding p-value $p_i$. Going through all pre-trained models, we obtained $m$ p-values: $\{p_1, p_2, \ldots, p_m\}$, and sort them in ascending order: $p_{(1)} \leq p_{(2)} \leq \cdots \leq p_{(m)}$. Then, we identify the test input $\mathbf{x}^*$ as an OOD sample if there exists an integer $1 \leq k \leq m$ such that $p_{(k)} \leq \frac{k}{m}(1 - \text{TPR}_0)$. Here 'TPR$_0$' is a predetermined TPR level of the ID data. In general, it is taken to be $95\%$. We call the proposed method Zoo-based OOD Detection Enhancement (ZODE) and present the details of ZODE in Algorithm 1. Next, we provide theoretical guarantees that Algorithm 1 can maintain the target TPR level on ID data.

**Theorem 1** *Suppose a pre-trained model zoo $\{\phi_1, \phi_2, \ldots, \phi_m\}$ is accessible and the score function is $S(x; \phi)$. Let TPR$_0 > 0.5$ be a predetermined TPR level for the ID Data. If the test input $\mathbf{x}^*$ is an ID sample that $\mathbf{x}^* \sim \mathcal{D}_{id}$ and $S(\mathbf{x}^*; \phi_i)$ is independent of $S(\mathbf{x}^*; \phi_j)$ for $\forall i \neq j$, then Algorithm 1 can identify $\mathbf{x}^*$ as an ID data with probability larger than $TPR_0$.*

**Remark.** Here we assume that $S(\mathbf{x}^*; \phi_i)$ is independent of $S(\mathbf{x}^*; \phi_j)$, which leads to the independence between $p_i$ and $p_j$ for $\forall i \neq j$. This assumption can hold if different pre-trained models learn completely different features. In this case, the model zoo haves the desired diversity. In practice, the pre-trained models can still be very diverse but different models may extract related features. Therefore, we report the empirical TPR of our method in Section 4. One can find that ZODE can still maintain the empirical TPR not less than the target level though the p-values may be related. The proof is postponed to Appendix A. In Appendix B, we analyze the detection power of Algorithm 1

---

**Algorithm 1** ZODE: Zoo-based OOD Detection Enhancement

---

**Require:** Training data $\{\mathbf{x}_i\}_{i=1}^n$, pre-trained model zoo $\{\phi_1, \ldots, \phi_m\}$, test sample $\mathbf{x}^*$, detection score $S(x; \phi)$, TPR level for ID data 'TPR$_0$';
 1: **Stage 1. Inference**
 2: Compute the score value of $S(\mathbf{x}_i, \phi_j)$, $\forall 1 \leq i \leq n$ and $\forall 1 \leq j \leq m$;
 3: **Stage 2. Testing**
 4: **for** $1 \leq j \leq m$ **do**
 5:     Estimate the p-value of $\mathbf{x}^*$ given $\phi_j$:

$$p_j = \frac{\#\{x_i : S(\mathbf{x}_i, \phi_j) \leq S(\mathbf{x}^*, f_j)\}}{m}$$

 6: **end for**
 7: **Stage 3. Ensemble**
 8: Sort $\{p_1, \ldots, p_m\}$ in ascending order: $\{p_{(1)}, \ldots, p_{(m)}\}$;
 9: **if** $\exists 1 \leq k \leq m$ such that $p_{(k)} \leq \frac{k}{m}(1 - \text{TPR}_0)$ **then**
10:     **return** $\mathbf{x}^*$ is an OOD sample;
11: **else**
12:     **return** $\mathbf{x}^*$ is an ID sample.
13: **end if**

---

as $m$ tends to infinity, which implies that FPR is guaranteed as the size of the model zoo increases. Appendix D presents more discussions about the ensemble scheme and compares the BH procedure with three baseline ensemble schemes.

**Computational complexity.** In Algorithm 1, we decompose ZODE into three stages: inference, testing, and ensemble. The inference stage requires computing the score of all validation samples for all pre-trained models, and its computational complexity is $m$ times that of post hoc OOD detection using a single pre-trained model since ZODE uses $m$ pre-trained models. However, the inference stage only needs the ID data and can be done before deploying the OOD detector. Therefore, its computational complexity does not increase the detection time when testing new inputs. In addition, the inference stage is only feed-forward and is easily parallelizable. Therefore, the computational burden is not heavy. In this work, all experiments can be done using one NVIDIA V100 GPU.

**Interpretability.** One of the benefits of ZODE is interpretability. If a test input is classified as an OOD sample, we can track which pre-models lead to this detection decision. At Step 9 of Algorithm 1, if $p_{(k)} \leq \frac{k}{m}(1 - \text{TPR}_0)$ and $p_{(j)} > \frac{j}{m}(1 - \text{TPR}_0)$, $\forall j > k$, then there are $k$ pre-trained models, corresponding to $p_{(1)}, \ldots, p_{(k)}$ respectively, that identify the test input as an OOD sample. In our experiments, we exploit this interpretability to find that there are OOD images that only one pre-trained model can detect. This implies that ZODE leverages the complementarity between all single-model detectors.

**Limitation.** The limitation of ZODE is that the testing stage takes up a lot of storage space. Post hoc OOD detection computes the score of all validation samples and selects a hard threshold by the quantile of the empirical distribution of the detection score. Therefore, post hoc OOD detection only passes the threshold from the inference stage to the testing stage. In Algorithm 1, the testing stage requires the score of validation samples to compute p-values. Therefore, the testing stage of ZODE takes up more storage space than post hoc OOD detection methods.

## 4 EXPERIMENTS

In this section, we demonstrate the effectiveness of our proposed method. First, we evaluate whether our model zoo and ensemble scheme can enhance OOD detectors. Second, we demonstrate that ZODE exploits the diversity of pre-trained models and leverages the complementarity between the single-model detectors to achieve superior performance. Finally, we show that our method can significantly improve the current SOTA results.

**Dataset**: We evaluate our proposed method on the CIFAR benchmarks. We use CIFAR10 (Krizhevsky et al., 2009) as the ID data and evaluate OOD detectors on six OOD datasets: SVHN

Table 1: **Results on CIFAR10.** Comparison with competitive OOD detection methods. The results of all competitors are from Sun et al. (2022). All values are percentages. ↓ indicates smaller values are better and vice versa.

| | | OOD Dataset | | | | | | | | | | | | |
| Method | | SVHN | | LSUN | | iSUN | | Texture | | Places365 | | Average | |
| | TPR | FPR↓ | AUC↑ | FPR↓ | AUC↑ | FPR↓ | AUC↑ | FPR↓ | AUC↑ | FPR↓ | AUC↑ | FPR↓ | AUC↑ |
|---|---|---|---|---|---|---|---|---|---|---|---|---|---|
| MSP | 95.00 | 59.66 | 91.25 | 45.21 | 93.80 | 54.57 | 92.12 | 66.45 | 88.50 | 62.46 | 88.64 | 57.67 | 90.86 |
| ODIN | 95.00 | 20.93 | 95.55 | 7.26 | 98.53 | 33.17 | 94.65 | 56.40 | 86.21 | 63.04 | 86.57 | 36.16 | 92.30 |
| Energy | 95.00 | 54.41 | 91.22 | 10.19 | 98.05 | 27.52 | 95.59 | 55.23 | 89.37 | 42.77 | 91.02 | 38.02 | 93.05 |
| GODIN | 95.00 | 15.51 | 96.60 | 4.90 | 99.07 | 34.03 | 94.94 | 46.91 | 89.69 | 62.63 | 87.31 | 32.80 | 93.52 |
| Mahalanobis | 95.00 | 9.24 | 97.80 | 67.73 | 73.61 | 6.02 | 98.63 | 23.21 | 92.91 | 83.50 | 69.56 | 37.94 | 86.50 |
| KNN | 95.00 | 24.53 | 95.69 | 25.29 | 95.96 | 25.55 | 95.26 | 27.57 | 94.71 | 50.90 | 89.14 | 30.77 | 94.15 |
| CSI | 95.00 | 37.38 | 94.69 | 5.88 | 98.86 | 10.36 | 98.01 | 28.85 | 94.87 | 38.31 | 93.04 | 24.16 | 95.89 |
| SSD+ | 95.00 | **1.51** | **99.68** | 6.09 | 98.48 | 33.60 | 95.16 | 12.98 | 97.70 | 28.41 | 94.72 | 16.52 | 97.15 |
| KNN+ | 95.00 | 2.42 | 99.52 | 1.78 | 99.48 | 20.06 | 96.74 | 8.09 | 98.56 | 23.02 | 95.36 | 11.07 | 97.93 |
| **ZODE**-MSP | 95.04 | 52.44 | 92.86 | 15.11 | 97.62 | 30.98 | 95.63 | 43.16 | 94.68 | 43.58 | 94.55 | 37.05 | 95.07 |
| **ZODE**-Energy | 95.07 | 50.05 | 92.26 | 3.12 | 99.29 | 16.03 | 97.09 | 37.34 | 95.14 | 19.52 | 96.95 | 25.21 | 96.15 |
| **ZODE**-Mahalanobis | 94.99 | 18.24 | 96.30 | 6.28 | 98.48 | 7.17 | 98.55 | 3.88 | 99.12 | 72.25 | 85.93 | 21.56 | 95.68 |
| **ZODE**-KNN | 94.96 | 2.12 | 99.43 | **1.50** | **99.61** | **5.48** | **98.70** | **0.16** | **99.88** | 9.91 | 97.99 | **3.83** | **99.12** |

(Netzer et al., 2011), LSUN (Yu et al., 2015), iSUN (Xu et al., 2015), Texture (Cimpoi et al., 2014), Places365 (Zhou et al., 2017), and CIFAR100 (Krizhevsky et al., 2009). We then consider more challenging benchmarks based on ImageNet, i.e., large-scale OOD detection tasks. The ID data is ImageNet-1K (Deng et al., 2009). We evaluate OOD detectors on four test datasets that are subset of : Places365 (Zhou et al., 2017), iNaturalist (Van Horn et al., 2018), SUN (Xiao et al., 2010), and Texture (Cimpoi et al., 2014) with different categories of each other.

**Metrics**: We evaluate OOD detection methods by the following three metrics: (1) the true positive rate of the ID samples (TPR); (2) the false positive rate of OOD samples when the true positive rate of the ID samples is about $95\%$ (FPR); (3) the area under the receiver operating characteristic curve (AUC). For single-model detectors, the hard threshold is determined by TPR $= 95\%$. Therefore, the first metric aims to check whether our ensemble scheme can maintain the TPR level close to $95\%$. FPR and AUC are often used in the literature to reflect the capabilities of OOD detectors. For the AUC metric, we use grid values of TPR ranging from 0 to 1 with a gap of 0.0005 and obtain the corresponding FPR to compute the area under the receiver operating characteristic curve.

**Enhanced OOD detection**: We consider three OOD detection methods: MSP (Hendrycks & Gimpel, 2017), Energy (Liu et al., 2020) and KNN (Sun et al., 2022). MSP is a simple baseline method that uses maximum softmax probabilities as the detection score. In some experiments, MSP can yield surprisingly good results when used on top of a large pre-trained model that has been fine-tuned on the ID data (Fort et al., 2021). The energy-based model (LeCun et al., 2006) maps a test input to a scalar that is higher for OOD samples and lower for the training data. Liu et al. (2020) proposes an energy score that uses the logits output by a pre-trained classifier. Sun et al. (2022) uses the feature distance between the test input and the $k$-th nearest ID sample and proposes a KNN-based detector. These three OOD detection methods represent three kinds of detectors based on probability, logit, and distance, respectively. We take them as the baseline methods and denote our enhanced methods by 'ZODE-MSP', 'ZODE-Energy', and 'ZODE-KNN' respectively.

## 4.1 Evaluation on CIFAR10 Benchmarks

**Model Zoo.** We build a model zoo with seven pre-trained models: ResNet18, ResNet34, ResNet50, ResNet101, ResNet152 (He et al., 2016), DenseNet (Huang et al., 2017) and ResNet18* (Sun et al., 2022). Here ResNet and DenseNet are two backbones routinely used in the literature on OOD detection. Therefore, we consider different architectures and use six models trained by cross-entropy loss. In addition, we also notice the effect of the loss function and introduce the model ResNet18* which is trained with contrastive loss. In summary, our model zoo contains diversity derived from different architectures and different training strategies.

**ZODE maintains TPR.** According to Section 3.1, one of the challenges of ensembling OOD detectors is to control the true positive rate of the ID data. Theorem 1 states that if different pre-trained models learn completely different features, ZODE can keep TPR close to the target level. In Table 1, we report the empirical TPR of ZODE, which is close to the target level $95\%$.

Table 2: **Results on CIFAR10.** We compare the ZODE-KNN detector with the single-model KNN detector. All values are percentages. ↓ indicates smaller values are better and vice versa.

| Method | TPR | SVHN FPR↓ | SVHN AUC↑ | LSUN FPR↓ | LSUN AUC | iSUN FPR↓ | iSUN AUC↑ | Texture FPR↓ | Texture AUC | Places365 FPR↓ | Places365 AUC↑ | Average FPR↓ | Average AUC↑ |
|---|---|---|---|---|---|---|---|---|---|---|---|---|---|
| ResNet18 | 95.00 | 27.97 | 95.49 | 18.50 | 96.84 | 24.68 | 95.52 | 26.74 | 94.97 | 47.95 | 90.02 | 29.17 | 94.57 |
| ResNet18* | 95.00 | 2.42 | **99.52** | 1.78 | 99.48 | 20.06 | 96.74 | 8.09 | 98.57 | 22.82 | 95.32 | 11.03 | 97.93 |
| ResNet34 | 95.00 | 26.53 | 95.85 | 10.22 | 98.39 | 29.45 | 95.15 | 31.65 | 94.53 | 36.59 | 92.75 | 26.89 | 95.33 |
| ResNet50 | 95.00 | 17.31 | 97.40 | 7.10 | 98.83 | 17.32 | 97.26 | 20.85 | 96.59 | 41.35 | 91.61 | 20.79 | 96.34 |
| ResNet101 | 95.00 | 25.73 | 96.12 | 6.65 | 98.90 | 19.84 | 96.80 | 18.42 | 96.89 | 40.57 | 92.15 | 22.24 | 96.17 |
| ResNet152 | 95.00 | 34.96 | 94.98 | 7.22 | 98.88 | 22.30 | 96.66 | 20.76 | 96.60 | 38.57 | 92.36 | 24.76 | 95.90 |
| DenseNet | 95.00 | 10.22 | 98.18 | 7.90 | 98.60 | 10.87 | 97.94 | 20.78 | 96.25 | 50.14 | 88.92 | 19.98 | 95.98 |
| **ZODE**-KNN | 94.96 | **2.12** | 99.43 | **1.50** | **99.61** | **5.48** | **98.70** | **0.16** | **99.88** | **9.91** | **97.99** | **3.83** | **99.12** |

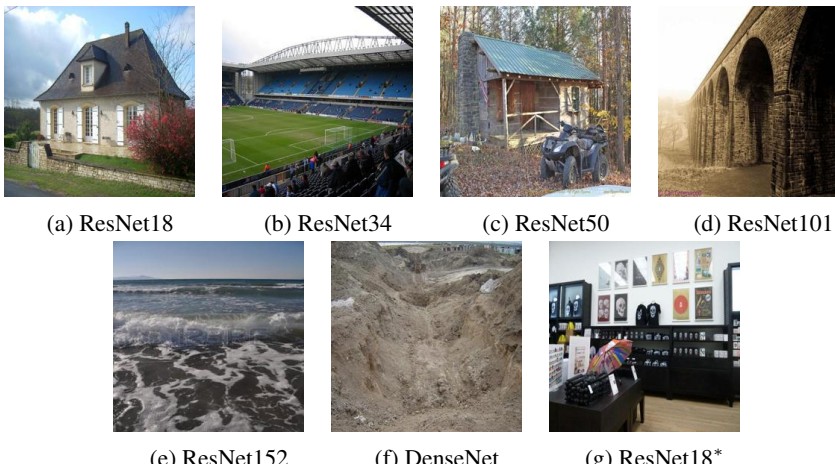

(a) ResNet18     (b) ResNet34     (c) ResNet50     (d) ResNet101

(e) ResNet152     (f) DenseNet     (g) ResNet18*

Figure 1: **Places365.** Example OOD images that only one single-model detector can identify.

**ZODE-KNN achieves superior performance.** We compare our method with competitive OOD detection methods, including MSP (Hendrycks & Gimpel, 2017), ODIN (Liang et al., 2018), Energy (Liu et al., 2020), GODIN (Hsu et al., 2020), Mahalanobis (Lee et al., 2018), KNN (Sun et al., 2022), CSI (Tack et al., 2020), SSD+ (Sehwag et al., 2020), as well as KNN+ (Sun et al., 2022). We cite the results of the competitors reported in Sun et al. (2022). For a fair comparison, we set $k = 50$ in the experiments of ZODE-KNN, which is the same as Sun et al. (2022). We can find that compared to the best baseline KNN+, ZODE-KNN reduces the FPR from $11.07\%$ to $3.83\%$, which significantly improves the relative detection accuracy by $65.40\%$. Note that ZODE-KNN significantly reduces FPR when OOD samples are drawn from iSUN, Texture, and Places365. For LSUN, ZODE-KNN slightly improves the performance of KNN+. In addition, SSD+ outperforms ZODE-KNN on SVHN. Overall, ZODE-KNN significantly improves the performance of existing methods on these five OOD datasets.

**ZODE achieves consistent improvements.** We consider three different kinds of OOD detection scores. MSP (Hendrycks & Gimpel, 2017) is based on the probabilities, Energy (Liu et al., 2020) uses the logits, and Mahalanobis (Lee et al., 2018) and KNN (Sun et al., 2022) quantify the distance in the embedding space. Then we compare them with the corresponding enhanced detectors: ZODE-MSP, ZODE-Energy, ZODE-Mahalanobis, and ZODE-KNN. For ZODE-MSP, ZODE-Energy, and ZODE–Mahalanobis, we use the same settings as Hendrycks & Gimpel (2017) and Liu et al. (2020). We find that ZODE-enhanced detectors consistently improve the performance of the corresponding baselines (Table 1).

**ZODE leverages the complementarity between the single-model detectors.** Table 2 reports the results of all single-model detectors derived from our model zoo and KNN score. It is easy to see that the ZODE-ensembled KNN detector significantly outperforms all single-model KNN detectors on LSUN, iSUN, Texture, and Places365. Compared with the best single-model baseline, ZODE reduces the FPR from $11.03\%$ to $3.83\%$, which significantly improves the relative detection accuracy by $65.28\%$. Moreover, ZODE improves the performance sharply on Texture and Places365. This

Table 3: **Results on CIFAR10 for CIFAR100 as OOD.** The results of GRAM and MaSF are from Haroush et al. (2022). We cite the results of SSD and SSD+ reported in (Sehwag et al., 2020). All values are percentages. ↓ indicates smaller values are better and vice versa.

(a) Comparison with baseline methods

| Method | TPR | FPR↓ | AUC↑ |
|---|---|---|---|
| GRAM | 95.00 | 51.00 | 83.30 |
| MaSF | 95.00 | 58.20 | 86.10 |
| SSD | 95.00 | 50.78 | 90.63 |
| SSD+ | 95.00 | 38.50 | 93.40 |
| KNN | 95.00 | 52.54 | 89.69 |
| KNN+ | 95.00 | 38.83 | 92.75 |
| **ZODE**-KNN | 94.96 | **18.29** | **97.12** |

(b) Ensembled vs Single-model

| Method | TPR | FPR↓ | AUC↑ |
|---|---|---|---|
| ResNet18 | 95.00 | 52.24 | 89.69 |
| ResNet18* | 95.00 | 38.83 | 92.75 |
| ResNet34 | 95.00 | 46.74 | 91.04 |
| ResNet50 | 95.00 | 47.14 | 90.64 |
| ResNet101 | 95.00 | 47.07 | 90.87 |
| ResNet152 | 95.00 | 47.72 | 90.84 |
| DenseNet | 95.00 | 49.43 | 89.80 |
| **ZODE**-KNN | 94.96 | **18.29** | **97.12** |

implies that the superior performance of ZODE does not fully come from any single-model detector. Therefore, our ensemble procedure works and is necessary for the improvements.

We further take Place365 as an example to illustrate that ZODE exploits the diversity of multiple pre-trained models. At step 9 of Algorithm 1, if $p_{(1)} \leq \frac{1}{m}(1 - \text{TPR}_0)$ and $p_{(j)} > \frac{j}{m}(1 - \text{TPR}_0)$, $\forall j \geq 2$, then there is only one pre-trained model that can help to identify the test input as an OOD sample. Figure 1 presents seven such images and each image corresponds to one pre-trained model in our model zoo.

**Evaluations on CIFAR10 vs CIFAR100.** We consider a challenging OOD detection task that identifies OOD samples drawn from CIFAR100 when the ID data is CIFAR10. Table 3a summarizes a detailed comparison with GRAM (Sastry & Oore, 2019), MaSF (Haroush et al., 2022), SSD (Sehwag et al., 2020), and KNN (Sun et al., 2022). Compared with the best baseline SSD+, ZODE reduces the FPR by 20.21%, which is a relative 52.49% improvement in detection power. The results in Table 3b clearly show that ZODE significantly outperforms the single-model-based KNN detectors and our ensemble scheme fully leverages the complementarity between the single-model detectors.

## 4.2 EVALUATION ON IMAGENET BENCHMARKS

**Model zoo and implementation details.** We use five pre-trained models to build a model zoo, consisting of models with different architectures and different pre-training strategies. The models are as follows: ResNet50* (Sun et al., 2022), semi-weakly supervised ResNeXt101 32x16d (Yalniz et al., 2019), Swinv2-B256, Swinv2-B384, and Swinv2-L256 (Liu et al., 2022). Significantly, resolutions of Swinv2-B256, Swinv2-B384, and Swinv2-L256 are 256x256, 256x256, and 384x384 respectively. ResNet50* is trained with SupCon loss (Khosla et al., 2020), which pulls points belonging to the same class together in the embedding space and separates samples from different classes. ResNeXt101 is pre-trained on Billion-scale images associated with meta information semantically relevant to ImageNet, which achieves 84.8% top-1 accuracy on ImageNet. The three Swinv2 models are pre-trained at higher resolution, and their top-1 accuracy on Imagenet all exceed 84%. In the following, we only report the results of ZODE-KNN based on the model zoo. The hyperparameter $\text{TPR}_0$ is taken to be 93.50%, which makes the empirical TPR of ZODE-KNN close to 95%. We use $k = 1000$ for ResNet50*, which is same as Sun et al. (2022). For the rest models, we selected $k$ from $\{100, 200, 500, 700, 800, 900, 1000, 3000, 5000\}$ that minimize the FPR.

**ZODE+KNN achieves superior performance.** In Table 4, we compare ZODE-KNN with competitive OOD detection methods, including MSP (Hendrycks & Gimpel, 2017), ODIN (Liang et al., 2018), Energy (Liu et al., 2020), GODIN (Hsu et al., 2020), Mahalanobis (Lee et al., 2018), KNN (Sun et al., 2022), SSD+ (Sehwag et al., 2020), as well as KNN+ (Sun et al., 2022). ZODE-KNN outperforms the best baseline KNN+ uniformly on all four OOD datasets, substantially reducing the average FPR from 38.47% to 28.10%, which achieves a relative 26.96% improvement in detection power. Especially when test datasets are iNaturalist and Textures, ZODE-KNN reduces the relative FPR by 83.40% and 70.61% respectively, which highlights the effectiveness of ZODE.

Table 4: **Results on ImageNet.** All results of the competitors are cited from Sun et al. (2022). Methods reported are all based on ID data only (ImageNet-1k). All values are percentages. ↓ indicates smaller values are better and vice versa.

| Method | TPR | OOD Dataset | | | | | | | | | |
| | | iNaturalist | | SUN | | Places | | Textures | | Average | |
| | | FPR↓ | AUC↑ | FPR↓ | AUC↑ | FPR↓ | AUC↑ | FPR↓ | AUC↑ | FPR↓ | AUC↑ |
| MSP | 95.00 | 54.99 | 87.74 | 70.83 | 80.86 | 73.99 | 79.76 | 68.00 | 79.61 | 66.95 | 81.99 |
| ODIN | 95.00 | 47.66 | 89.66 | 60.15 | 84.59 | 67.89 | 81.78 | 50.23 | 85.62 | 56.48 | 85.41 |
| Energy | 95.00 | 55.72 | 89.95 | 59.26 | 85.89 | 64.92 | 82.86 | 53.72 | 85.99 | 58.41 | 86.17 |
| GODIN | 95.00 | 61.91 | 85.40 | 60.83 | 85.60 | 63.70 | 83.81 | 77.85 | 73.27 | 66.07 | 82.02 |
| Mahalanobis | 95.00 | 97.00 | 52.65 | 98.50 | 42.41 | 98.40 | 41.79 | 55.80 | 85.01 | 87.43 | 55.47 |
| KNN | 95.00 | 59.00 | 86.47 | 68.82 | 80.72 | 76.28 | 75.76 | 11.77 | 97.07 | 53.97 | 85.01 |
| SSD+ | 95.00 | 57.16 | 87.77 | 78.23 | 73.10 | 81.19 | 70.97 | 36.37 | 88.52 | 63.24 | 80.09 |
| KNN+ | 95.00 | 30.18 | 94.89 | 48.99 | 88.63 | 59.15 | 84.71 | 15.55 | 95.40 | 38.47 | 90.91 |
| **ZODE**-KNN | 94.89 | **5.01** | **98.60** | **48.87** | **90.37** | **53.96** | **88.07** | **4.57** | **98.93** | **28.10** | **93.99** |

Table 5: **Results on ImageNet.** Comparison with single-model detectors and ZODE. All values are percentages. ↓ indicates smaller values are better and vice versa.

| Method | TPR | OOD Dataset | | | | | | | | | |
| | | iNaturalist | | SUN | | Places | | Textures | | Average | |
| | | FPR↓ | AUC↑ | FPR↓ | AUC↑ | FPR↓ | AUC↑ | FPR↓ | AUC↑ | FPR↓ | AUC↑ |
| ResNet50* | 95.00 | 30.18 | 94.89 | 48.99 | 88.63 | 59.15 | 84.71 | 15.55 | 95.40 | 38.47 | 90.91 |
| ResNext101 32x16 | 95.00 | 15.24 | 96.78 | 56.06 | 88.60 | 61.74 | 86.29 | 26.06 | 93.53 | 39.78 | 91.30 |
| Swinv2-B256 | 95.00 | 9.11 | 97.93 | 58.16 | 88.78 | 58.66 | 87.13 | 41.24 | 89.67 | 41.79 | 90.88 |
| Swinv2-B384 | 95.00 | 5.65 | 98.50 | 49.59 | 90.28 | **52.27** | **88.44** | 38.37 | 89.99 | 36.47 | 91.80 |
| Swinv2-L256 | 95.00 | 6.98 | 98.44 | 52.43 | 89.49 | 53.81 | 88.07 | 39.26 | 89.92 | 38.12 | 91.48 |
| **ZODE**-KNN | 94.89 | **5.01** | **98.60** | **48.87** | **90.37** | 53.96 | 88.07 | **4.57** | **98.93** | **28.10** | **93.99** |

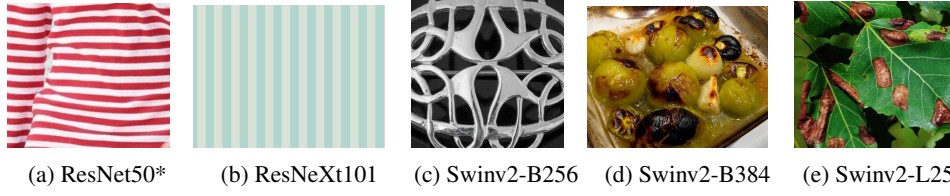

(a) ResNet50*  (b) ResNeXt101  (c) Swinv2-B256  (d) Swinv2-B384  (e) Swinv2-L256

Figure 2: **Textures.** Example OOD images that only one single-model detector can identify.

**ZODE combines the advantages of the single-model detectors.** In Table 5, we report the performance of every single-model detector derived from our model zoo. We highlight three trends: (1) ZODE-KNN outperforms the best single-model KNN detector with a relative 22.95% improvement in FPR. This implies that ZODE works in the ImageNet benchmarks and the ensemble scheme of ZODE-KNN is necessary for the improvements. (2) ZODE combines the advantages of single-model detectors. In Table 5, we can observe that ResNet50* and ResNeXt101 32x16 perform well on Textures, but underperform on iNaturalist, while the Swin models show the opposite performance. However, the ZODE-ensembled detector achieves strong and stable performance in all test datasets. (3) ZODE leverages the complementarity between the single-model detectors. Similar to the discussions in Figure 1, we find some images in Textures that can be successfully identified as OOD samples and the detection decision depends only on one single-model detector. Figure 1 presents five such images and each image corresponds to one pre-trained model in our model zoo.

## 5 CONCLUSION

In this paper, we exploit the diversity of multiple pre-trained models in a model zoo to improve the performance of post hoc OOD detection. We propose, ZODE, an efficient and fundamental ensemble scheme for combining multiple detection decisions. Extensive experiments show that ZODE can effectively solve the missed detection problem of single-model detectors by exploiting the complementarity of multiple detectors. We find that ZODE combined with the KNN detector (Sun et al., 2022) works very well. On a wide range of OOD detection benchmarks, ZODE-KNN significantly improves the current SOTA results.

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

## A  PROOF OF THEOREM 1

**Theorem.** *Suppose a pre-trained model zoo $\{\phi_1, \phi_2, \ldots, \phi_m\}$ is accessible and the score function is $S(x; \phi)$. Let $TPR_0 > 0.5$ be the target TPR level for the ID data. If the test input $\mathbf{x}^*$ is an ID sample that $\mathbf{x}^* \sim \mathcal{D}_{id}$ and $S(\mathbf{x}^*; \phi_i)$ is independent of $S(\mathbf{x}^*; \phi_j)$ for $\forall i \neq j$, then Algorithm 1 can identify $\mathbf{x}^*$ as an ID data with probability larger than $TPR_0$.*

**Proof.** Before proving the theorem, we state a useful lemma that provides the distribution of p-values under the ID distribution.

**Lemma 2** *Suppose the test input $\mathbf{x}^*$ is an ID sample that $\mathbf{x}^* \sim \mathcal{D}_{id}$ and the detection score $S(\mathbf{x}^*; \phi)$ is a continuous random variable. We write the p-value of $\mathbf{x}^*$ as*

$$p_0 = P\big(S(\mathbf{x}; \phi) \leq S(\mathbf{x}^*; \phi)\big|\mathbf{x} \sim \mathcal{D}_{id}\big).$$

*Then $p_0$ follows the uniform distribution $U[0, 1]$.*

**Proof.** Let $F_\phi(s)$ be the cumulative distribution function of $S(\mathbf{x}; \phi)$ with $\mathbf{x} \sim \mathcal{D}_{id}$. Then,

$$p_0 = \mathbb{P}\big(S(\mathbf{x}; \phi) \leq S(\mathbf{x}^*; \phi)\big|\mathbf{x} \sim \mathcal{D}_{id}\big) = F_\phi(S(\mathbf{x}^*; \phi)).$$

By the continuity of $S(\mathbf{x}^*; \phi)$ and Lemma 21.1 of Van der Vaart (2000), we have

$$
\begin{aligned}
\mathbb{P}(p_0 < \alpha) &= 1 - \mathbb{P}\big(F_\phi(S(\mathbf{x}^*; \phi)) \geq \alpha\big) \\
&= 1 - \mathbb{P}\big(S(\mathbf{x}^*; \phi) \geq F_\phi^{-1}(\alpha)\big) = F_\phi(F_\phi^{-1}(\alpha)) = \alpha.
\end{aligned}
$$

Hence $p_0$ follows the uniform distribution $U[0, 1]$.

□

According to Lemma 2, for any $\phi_i \in \mathcal{M}$,

$$p_i = \mathbb{P}\big(S(\mathbf{x};\phi_i) \leq S(\mathbf{x}^*;\phi_i)\big|\mathbf{x} \sim \mathcal{D}_{id}\big) \ \sim \ U[0,1],$$

and the density function of $p_i$ is

$$f_{p_i}(x) = \begin{cases} 1 & x \in [0,1]; \\ 0 & \text{otherwise.} \end{cases}$$

Then, the joint probability density of the ordered values $p_{(1)}, p_{(2)}, ..., p_{(m)}$ is

$$f_{p_{(1)},p_{(2)},...,p_{(m)}}(x_1, x_2, ..., x_m) = m! \prod_{i=1}^{S} f_{p_i}(x_i) = m!$$

We denote $\alpha = 1 - \text{TPR}_0$ and define an event $\mathcal{E}$: $\forall\, 1 \leq j \leq m,\ p_{(j)} \geq \frac{j}{m}\alpha$. Then we have

$$\begin{aligned}
\mathbb{P}(\mathcal{E}|\mathbf{x}^* \sim \mathcal{D}_{id}) &= \int_{\frac{m}{m}\alpha}^{1} \cdots \int_{\frac{2}{m}\alpha}^{1} \int_{\frac{1}{m}\alpha}^{1} f_{p_{(1)},\cdots,p_{(m)}}(x_1, x_2, \cdots, x_m)\mathrm{d}x_1 \mathrm{d}x_2 \cdots \mathrm{d}x_m \\
&= m!(1 - \frac{1}{m}\alpha)(1 - \frac{2}{m}\alpha)...(1 - \frac{m}{m}\alpha).
\end{aligned} \tag{4}$$

Next, we prove that for any $m \geq 1$ and $\alpha \leq 0.5$,

$$m!(1 - \frac{1}{m}\alpha)(1 - \frac{2}{m}\alpha)...(1 - \frac{m}{m}\alpha) \geq 1 - \alpha \tag{5}$$

It is easy to see that Eq. (5) holds when $m = 1$. Suppose Eq. (5) holds for $m = m_0$. Then for $m = m_0 + 1$, we have

$$\begin{aligned}
&(m_0 + 1)!(1 - \frac{1}{m_0 + 1}\alpha)(1 - \frac{2}{m_0 + 1}\alpha)...(1 - \frac{m_0 + 1}{m_0 + 1}\alpha) \\
&\geq (m_0 + 1)!(1 - \frac{1}{m_0}\alpha)(1 - \frac{2}{m_0}\alpha)...(1 - \frac{m_0}{m_0}\alpha)(1 - \frac{m_0 + 1}{m_0 + 1}\alpha) \\
&\geq (m_0 + 1)(1 - \alpha)^2 \geq 1 - \alpha,
\end{aligned} \tag{6}$$

which implies that Eq.(5) also holds for $m = m_0 + 1$. Hence, the proof is finished.

□

## B  THE POWER OF ALGORITHM 1

In this section, we study the FPR of Algorithm 1 from the asymptotic perspective, i.e., $m$ tends to infinity. According to Section 3.2 and Appendix A, the p-value of an ID sample follows the uniform distribution $U[0,1]$. Also, if a pre-trained model fails to identify OOD samples, the p-value derived from the model follows the uniform distribution. For an OOD dataset, we assume that there is a fixed proportion $\pi$ of pre-trained models that can recognize the OOD data points. We call these models active models and denote the set of active models as $\mathcal{A}$. Then pre-trained models that fail to identify OOD samples belong to the set $\mathcal{A}^c = \mathcal{M} - \mathcal{A}$. We have, for any $0 \leq u \leq 1$,

$$\mathbb{P}(p_j \leq u|\phi_j \in \mathcal{A}^c) = u \quad \text{and} \quad \mathbb{P}(p_j \leq u|\phi_j \in \mathcal{A}) = G(u),$$

where $G(u)$ is a cumulative distribution function different from that of the uniform distribution $U[0,1]$. Therefore, the p-values of the OOD dataset are sampled from a mixture model with a cumulative distribution function:

$$F(u) = (1 - \pi)u + \pi G(u).$$

Let $k$ be the number of pre-trained models that classify the OOD input as an OOD sample. That is $p_{(k)} \leq \frac{k}{m}\alpha$ and $p_{(j)} > \frac{j}{m}\alpha, \forall j > k$ with $\alpha = 1 - \text{TPR}_0$. Table 6 summarizes the OOD detection

Table 6: **OOD detection.**

|  |  | Truth | | |
|---|---|---|---|---|
|  |  | ID | OOD | |
| Detect | ID | U | T | |
|  | OOD | V | S | $k$ |
|  |  | $m_0$ | $m_1$ | $m$ |

result with Algorithm 1. If $S \geq 1$, Algorithm 1 successfully detects an OOD sample. Next, we study the average power:

$$\mathbb{E}\Big(\frac{S}{m_1}\Big) = \mathbb{E}\Big(\frac{k \times \frac{S}{k}}{m_1}\Big) = \mathbb{E}\Big(\frac{\frac{k}{m} \times (1 - \frac{V}{k})}{\frac{m_1}{m}}\Big).$$

According to Chi (2007), $\frac{k}{m}$ converges to a positive value $p_*(\alpha, F)$ as $m \to \infty$, which serves as the limit of the proportion of rejected p-values. By Theorem 1 and its lemma in Benjamini & Hochberg (1995),

$$\mathbb{E}\Big(1 - \frac{V}{k}\Big) \geq 1 - \frac{m_0}{m}\alpha = 1 - (1 - \pi)\alpha.$$

Therefore, if $m$ is sufficiently large, then we have

$$\mathbb{E}\Big(\frac{S}{m_1}\Big) \geq \frac{p_*(\alpha, F)(1 - (1 - \pi)\alpha)}{\pi} \geq \frac{1}{\pi m} = \frac{1}{m_1}.$$

This implies that if $m$ is sufficiently large, $S$ is greater or equal to 1 with high probability. In other words, if the number of pre-trained models is sufficiently large, Algorithm 1 can identify OOD samples with high probability.

## C   USING P-VALUE FOR OOD DETECTION

In OOD detection, the p-value is a probability measure that quantifies how extreme the observed score is when the input is an ID sample (Cai & Koutsoukos, 2020; Morningstar et al., 2021; Haroush et al., 2022; Bergamin et al., 2022; Magesh et al., 2022; Kaur et al., 2022). Given a test sample $\mathbf{x}^*$, the lower value of $S(\mathbf{x}^*; \phi)$, the more likely $\mathbf{x}^*$ is not drawn from the training distribution. Hence, the p-value of $\mathbf{x}^*$ is the probability that $S(\mathbf{x}; \phi)$ is less than $S(\mathbf{x}^*; \phi)$ under the ID distribution:

$$\text{P-value of } \mathbf{x}^* = \mathbb{P}\big(S(\mathbf{x}; \phi) \leq S(\mathbf{x}^*; \phi)\big|\mathbf{x} \sim \mathcal{D}_{id}\big).$$

In practice, **using the p-value is equivalent to using the hard threshold** $S(\mathbf{x}^*) < \lambda$. We denote $\{(\mathbf{x}_i, \mathbf{y}_i)\}_{i=1}^n$ as validation data sampled from the ID distribution $\mathcal{P}_{id}$ and sort their detection score in ascending order:

$$S(\mathbf{x}_{(1)}; \phi) \leq S(\mathbf{x}_{(2)}; \phi) \leq \cdots \leq S(\mathbf{x}_{(n)}; \phi).$$

In post hoc OOD detection, the threshold $\lambda_\phi$ is determined by correctly classifying 95% of validation data as ID samples, i.e., TPR is at least 95%. Therefore,

$$S(\mathbf{x}_{(\lfloor 0.05n \rfloor)}; \phi) \leq \lambda_\phi \leq S(\mathbf{x}_{(\lfloor 0.05n \rfloor + 1)}; \phi),$$

where $\lfloor \cdot \rfloor$ is the floor function. On the other hand, the p-value of $\mathbf{x}^*$ less than 0.05 implies that

$$P\big(S(\mathbf{x}; \phi) \leq S(\mathbf{x}^*; \phi)\big|\mathbf{x} \sim \hat{\mathcal{D}}_{id}\big) \approx 0.05 \;\Rightarrow\; S(\mathbf{x}^*; \phi) \lessapprox S(\mathbf{x}_{(\lfloor 0.05n \rfloor + 1)}; \phi),$$

where $\hat{\mathcal{D}}_{id}$ is the empirical distribution of $\{\mathbf{x}_i\}_{i=1}^n$. Therefore, when the sample size $n$ is sufficiently large, the OOD region derived from the critical value $\{\mathbf{x} : S(\mathbf{x}; \phi) < \lambda_\phi\}$ is the same as the OOD region determined by the p-value $\{\mathbf{x} : \text{P-value of } \mathbf{x} < 0.05\}$.

Table 7: **Results on CIFAR10.** Comparison with three baseline ensemble schemes. All values are percentages. ↓ indicates smaller values are better and vice versa.

| Method | | OOD Dataset | | | | | | | | | | | | |
| | | **SVHN** | | **LSUN** | | **iSUN** | | **Texture** | | **Places365** | | **Average** | |
| | TPR | FPR↓ | AUC↑ | FPR↓ | AUC↑ | FPR↓ | AUC↑ | FPR↓ | AUC↑ | FPR↓ | AUC↑ | FPR↓ | AUC↑ |
|---|---|---|---|---|---|---|---|---|---|---|---|---|---|
| Naive | 69.89 | 0.16 | 99.18 | 0.12 | 99.39 | 0.53 | 98.21 | 0.00 | 99.74 | 0.14 | 97.36 | 0.95 | 98.78 |
| Average | 100.00 | 19.21 | 99.98 | 6.92 | 100.00 | 20.04 | 99.98 | 23.69 | 100.00 | 69.76 | 99.97 | 27.92 | 99.98 |
| Voting | 99.64 | 5.43 | 99.82 | 2.46 | 99.95 | 6.68 | 99.82 | 0.39 | 99.99 | 8.99 | 99.86 | 4.79 | 99.89 |
| BH | 94.96 | 2.12 | 99.43 | 1.50 | 99.61 | 5.48 | 98.70 | 0.16 | 99.88 | 9.91 | 97.99 | 3.83 | 99.12 |

Table 8: **Results on ImageNet.** Comparison with three baseline ensemble schemes. All values are percentages. ↓ indicates smaller values are better and vice versa.

| Method | | OOD Dataset | | | | | | | | | |
| | | **iNaturalist** | | **SUN** | | **Places** | | **Textures** | | **Average** | |
| | TPR | FPR↓ | AUC↑ | FPR↓ | AUC↑ | FPR↓ | AUC↑ | FPR↓ | AUC↑ | FPR↓ | AUC↑ |
|---|---|---|---|---|---|---|---|---|---|---|---|
| Naive | 88.53 | 2.00 | 98.54 | 29.53 | 89.60 | 36.72 | 86.93 | 1.45 | 99.00 | 17.43 | 93.52 |
| Average | 97.41 | 11.64 | 98.57 | 58.21 | 91.43 | 61.40 | 89.13 | 44.77 | 93.55 | 44.05 | 93.15 |
| Voting | 93.59 | 3.99 | 98.66 | 43.16 | 90.26 | 48.18 | 87.98 | 11.74 | 97.14 | 26.76 | 93.51 |
| BH | 94.89 | 5.01 | 98.60 | 48.87 | 90.37 | 53.96 | 88.07 | 4.57 | 98.93 | 28.10 | 93.99 |

# D    USING BH PROCEDURE FOR ENSEMBLE

In this work, we use the Benjamini–Hochberg procedure (Benjamini & Hochberg, 1995) to ensemble multiple detection decisions (p-values) to exploit the diversity of pre-trained models. Please see Steps 7 - 13 in Algorithm 1. Recall that $TPR_0$ is the target TPR level and $\alpha = 1 - TPR_0$. According to Theorem 1, this procedure controls the true positive rate of ID data at a level greater than $TPR_0$ by assuming independent p-values. Benjamini & Yekutieli (2001) points out that the procedure is also available for cases with certain types of positively related p-values.

In this section, we consider three ensemble schemes, Naive, Average, and Voting, as competitors to illustrate the superior of the BH procedure. Here 'Naive' represents the naive ensemble scheme in Eq.(2), 'Average' refers to the ensemble scheme that identifies a test input as an OOD sample if the average p-value is smaller than $\alpha$, and 'Voting' is the majority voting that identifies a test input as an OOD sample if more than half of the p-values are less than $\alpha$. The score function is the KNN score (Sun et al., 2022) with $k = 50$ and $TPR_0$ is $95\%$. The results are presented in Table 7 and 8. We can find that the empirical TPR of our method is well-controlled and is close to the target TPR level. The naive ensemble scheme cannot maintain the target TPR level. Therefore its low FPR is unreliable. This observation is consistent with our theoretical results in Section 3.1. The average ensemble scheme can maintain its empirical TPR larger than $95\%$ but its FPR is very large. Overall, the voting ensemble is comparable to our scheme. Its TPR on ImageNet is a bit out of control.

# E    INFLUENCE OF HYPERPARAMETER

In Secction 4, we find that the ZODE-KNN detector works very well and significantly improves the best baseline method on a wide range of OOD detection benchmarks. The KNN detector (Sun et al., 2022) uses the distance between the test input and the $k$-th nearest ID sample as the detection score. Here $k$ is a hyperparameter that needs to be predetermined. In this section, we use CIFAR10 to investigate the effect of the choice of $k$ on ZODE. We consider $k = 1$ and $k = 50$, and report the results in Table 9. One can find that the choice of k affects the performance of ZODE-KNN. But the influence is not significant compared to the improvements of ZODE-KNN. Both ZODE-KNN(k=1) and ZODE-KNN(k=50) significantly outperform the best baseline methods. In Table 10, we report the detailed comparison between the ZODE-ensembled KNN detector and the single-model KNN detectors derived from our model zoo. We observe a similar phenomenon in that the choice of k affects the performance of both ensembled and single-model detectors, while the influence is not significant compared to the improvement caused by the ensemble scheme.

Table 9: **Results on CIFAR10.** Comparison with competitive OOD detection methods. The results of all competitors are from Sun et al. (2022). All values are percentages. ↓ indicates smaller values are better and vice versa.

| Method | TPR | SVHN | | LSUN | | iSUN | | Texture | | Places365 | | Average | |
|---|---|---|---|---|---|---|---|---|---|---|---|---|---|
| | | FPR↓ | AUC↑ | FPR↓ | AUC↑ | FPR↓ | AUC↑ | FPR↓ | AUC↑ | FPR↓ | AUC↑ | FPR↓ | AUC↑ |
| MSP | 95.00 | 59.66 | 91.25 | 45.21 | 93.80 | 54.57 | 92.12 | 66.45 | 88.50 | 62.46 | 88.64 | 57.67 | 90.86 |
| ODIN | 95.00 | 20.93 | 95.55 | 7.26 | 98.53 | 33.17 | 94.65 | 56.40 | 86.21 | 63.04 | 86.57 | 36.16 | 92.30 |
| Energy | 95.00 | 54.41 | 91.22 | 10.19 | 98.05 | 27.52 | 95.59 | 55.23 | 89.37 | 42.77 | 91.02 | 38.02 | 93.05 |
| GODIN | 95.00 | 15.51 | 96.60 | 4.90 | 99.07 | 34.03 | 94.94 | 46.91 | 89.69 | 62.63 | 87.31 | 32.80 | 93.52 |
| Mahalanobis | 95.00 | 9.24 | 97.80 | 67.73 | 73.61 | 6.02 | 98.63 | 23.21 | 92.91 | 83.50 | 69.56 | 37.94 | 86.50 |
| KNN | 95.00 | 24.53 | 95.69 | 25.29 | 95.96 | 25.55 | 95.26 | 27.57 | 94.71 | 50.90 | 89.14 | 30.77 | 94.15 |
| CSI | 95.00 | 37.38 | 94.69 | 5.88 | 98.86 | 10.36 | 98.01 | 28.85 | 94.87 | 38.31 | 93.04 | 24.16 | 95.89 |
| SSD+ | 95.00 | **1.51** | **99.68** | 6.09 | 98.48 | 33.60 | 95.16 | 12.98 | 97.70 | 28.41 | 94.72 | 16.52 | 97.15 |
| KNN+ | 95.00 | 2.42 | 99.52 | 1.78 | 99.48 | 20.06 | 96.74 | 8.09 | 98.56 | 23.02 | 95.36 | 11.07 | 97.93 |
| **ZODE**-KNN(k=1) | 95.00 | 2.60 | 99.42 | 2.34 | 99.46 | 7.29 | 98.47 | 0.44 | 99.73 | 11.79 | 97.77 | 4.89 | 98.97 |
| **ZODE**-KNN(k=50) | 94.96 | 2.12 | 99.43 | **1.50** | **99.61** | **5.48** | **98.70** | **0.16** | **99.88** | 9.91 | 97.99 | 3.83 | **99.12** |

Table 10: **Results on CIFAR10**. Comparison with single-model detectors at different $k$ levels. Setting $k = 1$ and $k = 50$ respectively, we compare the performance with single-model detectors and ZODE.

| K | Method | TPR | SVHN | | LSUN | | iSUN | | Texture | | Places365 | | Average | |
|---|---|---|---|---|---|---|---|---|---|---|---|---|---|---|
| | | | FPR↓ | AUC↑ | FPR↓ | AUC | FPR↓ | AUC↑ | FPR↓ | AUC | FPR↓ | AUC↑ | FPR↓ | AUC↑ |
| $k=1$ | ResNet18 | 95.00 | 50.34 | 91.45 | 25.44 | 95.67 | 30.53 | 94.71 | 33.44 | 93.69 | 48.40 | 90.05 | 37.63 | 93.11 |
| | ResNet18* | 95.00 | **2.16** | **99.56** | 4.45 | 98.78 | 21.99 | 96.56 | 9.88 | 98.34 | 23.87 | 95.38 | 12.47 | 97.72 |
| | ResNet34 | 95.00 | 28.09 | 95.55 | 11.80 | 98.13 | 30.90 | 94.97 | 32.19 | 94.24 | 38.67 | 92.41 | 28.19 | 95.06 |
| | ResNet50 | 95.00 | 22.37 | 96.25 | 7.15 | 98.67 | 18.40 | 96.90 | 23.63 | 95.72 | 41.00 | 91.64 | 22.51 | 95.84 |
| | ResNet101 | 95.00 | 25.45 | 95.96 | 8.46 | 98.62 | 20.92 | 96.54 | 21.03 | 96.31 | 41.07 | 92.18 | 23.39 | 95.92 |
| | ResNet152 | 95.00 | 33.79 | 94.16 | 8.48 | 98.63 | 19.06 | 96.93 | 20.62 | 96.50 | 37.97 | 92.52 | 23.98 | 95.75 |
| | DensNet | 95.00 | 13.46 | 97.76 | 9.41 | 98.31 | 13.70 | 97.58 | 24.29 | 95.46 | 51.49 | 89.07 | 22.47 | 95.64 |
| | **ZODE**-KNN | 95.00 | 2.60 | 99.42 | **2.34** | **99.46** | **7.29** | **98.47** | **0.44** | **99.73** | **11.79** | **97.77** | **4.89** | **98.97** |
| $k=50$ | ResNet18 | 95.00 | 27.97 | 95.49 | 18.50 | 96.84 | 24.68 | 95.52 | 26.74 | 94.97 | 47.95 | 90.02 | 29.17 | 94.57 |
| | ResNet18* | 95.00 | 2.42 | **99.52** | 1.78 | 99.48 | 20.06 | 96.74 | 8.09 | 98.57 | 22.82 | 95.32 | 11.03 | 97.93 |
| | ResNet34 | 95.00 | 26.53 | 95.85 | 10.22 | 98.39 | 29.45 | 95.15 | 31.65 | 94.53 | 36.59 | 92.75 | 26.89 | 95.33 |
| | ResNet50 | 95.00 | 17.31 | 97.40 | 7.10 | 98.83 | 17.32 | 97.26 | 20.85 | 96.59 | 41.35 | 91.61 | 20.79 | 96.34 |
| | ResNet101 | 95.00 | 25.73 | 96.12 | 6.65 | 98.90 | 19.84 | 96.80 | 18.42 | 96.89 | 40.57 | 92.15 | 22.24 | 96.17 |
| | ResNet152 | 95.00 | 34.96 | 94.98 | 7.22 | 98.88 | 22.30 | 96.66 | 20.76 | 96.60 | 38.57 | 92.36 | 24.76 | 95.90 |
| | DensNet | 95.00 | 10.22 | 98.18 | 7.90 | 98.60 | 10.87 | 97.94 | 20.78 | 96.25 | 50.14 | 88.92 | 19.98 | 95.98 |
| | **ZODE**-KNN | 94.96 | **2.12** | 99.43 | **1.50** | **99.61** | **5.48** | **98.70** | **0.16** | **99.88** | 9.91 | 97.99 | 3.83 | 99.12 |

# F  MORE RELATED WORK

**Score function for OOD detection.** Many recent works have proposed to use models pre-trained with multi-class classification tasks to do OOD detection. A natural idea to distinguish OOD samples is to use the softmax confidence score based on the classifier output and a normalized probability vector. However, neural networks might give an OOD sample a high confidence prediction (Guo et al., 2017; Hein et al., 2019). To deal with this issue, several studies have improved scoring methods and also proposed new scoring measures, such as OpenMax score (Bendale & Boult, 2015), maximum softmax probability (Hendrycks & Gimpel, 2017), ODIN score (Liang et al., 2018), deep ensembles (Lakshminarayanan et al., 2017), Mahalanobis distance-based score (Lee et al., 2018), Gram matrix (Sastry & Oore, 2019), energy score (Liu et al., 2020), (Lin et al., 2021), activation rectification (ReAct) (Sun et al., 2021), gradient-based score (GradNorm) (Huang et al., 2021), and ViM score (Wang et al., 2022). In this work, we consider KNN score Sun et al. (2022) that employs the non-parametric nearest neighbor approach to quantify the distance between the test input and the training data.

**Model-training strategies for OOD detection.** Besides proposing efficient score functions, another area of research focuses on choosing strategies for training deep models, including generative adversarial networks (Schlegl et al., 2017), convolutional network (Sabokrou et al., 2018) and transfer representation-learning (Andrews et al., 2016). These classical neural network architectures, such as ResNet (He et al., 2016), DenseNet (Huang et al., 2017) and Swin (Liu et al., 2021) etc, show different effects in downstream tasks when trained with different training strategies or regularization methods (Ericsson et al., 2021). For example, utilizing auxiliary OOD training data like artificially synthesized data from GANs (Lee et al., 2017) and unlabeled data for outlier exposure

(Hendrycks et al., 2018) can explicitly regularize the model. Another method to fine-tune the model is to modify the loss function or use auxiliary objectives. Some loss functions encourage the predictive distribution of OOD sample toward uniform distribution (Lee et al., 2017; Hendrycks et al., 2018), and also some by adding contrastive loss (Winkens et al., 2020), margin loss (Vyas et al., 2018) or objective function of adversarial learning (Biggio & Roli, 2018; Miller et al., 2020; Chalapathy & Chawla, 2019) can force models to learn more high-level, task-agnostic, comprehensive features from the training dataset, to enable it robust enough for downstream tasks with various distribution shifts. These works either modify the neural network and objective function or require additional data, at the cost of computational cost.

