# OpenReview forum: "Boosting Out-of-Distribution Detection with Multiple Pre-trained Models "
_ICLR.cc/2023/Conference — Submitted to ICLR 2023_

### Official Review · Reviewer_FQTf · 2022-10-24

**Confidence:** 4
**Correctness:** 3
**Technical Novelty And Significance:** 3
**Empirical Novelty And Significance:** 2
**Recommendation:** 5

**Clarity, Quality, Novelty And Reproducibility:**

Writing is clear and well-written in general. The ensemble strategy is not novel, but the method to ensemble them, estimating and thresholding p-value might be novel. They also provided some theoretical analysis on it. I believe this work is reproducible with the provided code.

**Details Of Ethics Concerns:**

Nothing special.

**Strength And Weaknesses:**

Strengths

+ Theoretical analysis on the p-value looks interesting.

+ The proposed method clearly outperforms other methods in most cases.

Weaknesses

- No ablation study on the ensembling strategy. For example, authors could simply average the detection score or do majority voting and take them as baselines.

- The comparison is not fair. Model ensembling is a kind of guaranteed strategy to improve the performance in machine learning. Even authors showed consistently good performance, empirical contribution is somewhat limited in that sense. Rather, comparison with other (baseline) ensemble strategies should be fair, but missed.

- Is there a reason why the proposed method is combined with MSP, Energy, and KNN only? I wonder if the rank of methods holds, e.g., if ZODE-Mahalanobis is on average worse than ZODE-KNN. Authors could add more results in the appendix.

- Typo: DensNet @ Table 2

**Summary Of The Paper:**

This paper proposes to ensemble multiple pre-trained models with different architecture by estimating and thresholding p-value for out-of-distribution detection. Experimental results show the effectiveness of the proposed method.


**Summary Of The Review:**

I am mostly happy with this paper, except for missing ablation study on the choice of ensembling strategy. Again, I believe ensemble method almost always improves the performance of machine learning models, so comparison with other ensemble methods should be more reasonable than standalone models. Please answer my concerns above.


**post-rebuttal**

The additional experiments for comparison with simple ensemble methods addressed my concern, but it turned out that the performance of the proposed method is not so better than baseline ensemble methods, in terms of AUC. Ensembling based on the p-value and corresponding analysis are interesting, but by looking at the performance, I can't see the reason why we need such a sophisticated ensemble strategy.
At this point, I decided not to change my initial rating.

By the way, authors claim that TPR is "out of control" and show FPR under somewhat random TPR for other ensemble methods, which I do not fully understand what they actually mean by. Maybe authors observed discrete changes on TPR?

It seems authors also count the fact that their work is first to take advantage of "(publicly available) model zoo" for OOD detection as the main contribution, but by looking at other reviews, it seems other reviewers do not care about it much?

---

> ### Author Response · Authors · 2022-11-13
> **Reply to Reviewer FQTf (Part I)**
>
> We thank you for your time and interesting suggestion. We have conducted an ablation study in light of your recommendation.
>
> ___
>
> > **Q1.** No ablation study on the ensembling strategy. For example, authors could simply average the detection score or do majority voting and take them as baselines.
>
> **Ans:** Thank you for the comments. To the best of our knowledge, there are no baseline methods focused on the ensembled OOD detector in the model zoo setting. Therefore, in our ablation studies (Table 2, Table 3b, Table 5), we validate the effectiveness of our ensemble method and do not consider the comparison with other ensemble schemes. Here we conduct the ablation study by comparing four baseline ensemble schemes: **BH**, **Naive**, **Average**, and **Voting**.
>
> **BH:** The BH correction scheme we used in this work.
>
> **Naive:** If one single-model detector identifies the test input as an OOD sample, then we classify the test input as an OOD sample.
>
> **Average:** We use the mean of p-values as the detection score.
>
> **Voting:** Majority voting.
>
> The score function is KNN and the target TPR is taken to be 95%. We compare our ensemble scheme with the other three baseline methods on CIFAR10 and ImageNest benchmarks. The results are reported below. We will add the detailed results in the revised version.
>
> We can find that the TPR of our method is well-controlled and is close to the target TPR level. The naive ensemble scheme cannot maintain the target TPR level. Therefore its low FPR is unreliable. This observation is consistent with our theoretical results in Section 3.1. The average ensemble scheme can maintain its empirical TPR larger than 95% but its FPR is very large. Overall, the voting ensemble is comparable to our scheme. Its TPR on ImageNet is a bit out of control.
>
> **The results on CIFAR10 benchmarks：**
>
> |         | TPR   | mean FPR | mean AUC |
> | ------- | ----- | -------- | -------- |
> | BH(Our) | 94.96 | 3.83     | 99.12    |
> | Naive   | 69.89 | 0.95     | 98.78    |
> | Average | 100   | 27.92    | 99.98    |
> | Voting  | 99.64 | 4.79     | 99.89    |
>
> **The results on ImageNet benchmarks：**
>
> |         | TPR   | mean FPR | mean AUC |
> | ------- | ----- | -------- | -------- |
> | BH(Our) | 94.89 | 28.10    | 93.99    |
> | Naive   | 88.53 | 17.43    | 93.52    |
> | Average | 97.41 | 44.05    | 93.15    |
> | Voting  | 93.59 | 26.76    | 93.51    |
>
> ___
>
> > **Q2.** The comparison is not fair. Model ensembling is a kind of guaranteed strategy to improve the performance in machine learning. Even authors showed consistently good performance, empirical contribution is somewhat limited in that sense. Rather, comparison with other (baseline) ensemble strategies should be fair, but missed.
>
> **Ans:** In this work, we use a model zoo to boost the performance of OOD detection and name our proposed method Zoo-based OOD Detection Enhancement (ZODE). Our results (Table 1, Table 3a, Table 4) clearly show the power of the model zoo. Furthermore, our experiments validate that the proposed method can leverage the diversity of the model zoo to consistently improve the OOD detection methods. To the best of our knowledge, we are the first to systematically study using a model zoo to boost OOD detection. In the model zoo literature, there are no baseline methods on the same track as our problem.  Therefore, we focus on the effectiveness of the ensemble scheme.
>
> On the other hand, we have checked each pre-trained model in the model zoo (Table 2, Table 3b, Table 5). We can find that the ensembled detector can significantly outperform all single-model detectors. This implies that no pre-trained model can dominate the performance of the ensembled detector. Therefore, in our experiments, there does not exist the case: our model zoo contains one high-quality pre-trained model which dominates the improvement of the ensemble. This implies the effectiveness of the ensemble. In the revised version, we will add more experiments to empirically compare our ensemble scheme with the naive ensemble, average score, and majority voting.
> ___

---

> > ### Author Response · Authors · 2022-11-13
> > **Reply to Reviewer FQTf (Part II)**
> >
> > ___
> >
> > > **Q3.** Is there a reason why the proposed method is combined with MSP, Energy, and KNN only? I wonder if the rank of methods holds, e.g., if ZODE-Mahalanobis is on average worse than ZODE-KNN. Authors could add more results in the appendix.
> >
> > **Ans:** In this work, we consider three score functions: MSP [1], Energy [2], and KNN [3]. The MSP score considers the softmax output. The energy score uses the logits. The KNN score is derived from the embedding distance. On CIFAR10 benchmarks, we report the results of MSP vs ZODE-MSP, Energy vs ZODE-Energy, and KNN vs ZODE-KNN (Table 1). We find that ZODE-enhanced detectors consistently improve the performance of the corresponding baselines. Here we present the mean results of the Mahalanobis score. In the revised version, we will present the detailed results.
> >
> > | CIFAR10          | TPR   | mean FPR | mean AUC |
> > | ---------------- | ----- | -------- | -------- |
> > | MSP              | 95.00 | 57.67    | 90.86    |
> > | Energy           | 95.00 | 38.02    | 93.05    |
> > | Mahalanobis      | 95.00 | 37.94    | 86.50    |
> > | KNN              | 95.00 | 30.77    | 94.15    |
> > | KNN+             | 95.00 | 11.07    | 97.93    |
> > | ZODE-MSP         | 95.04 | 37.05    | 95.07    |
> > | ZODE-Energy      | 95.07 | 25.21    | 96.15    |
> > | ZODE-Mahalanobis | 94.99 | 21.56    | 95.68    |
> > | ZODE-KNN         | 94.96 | 3.83     | 99.12    |
> >
> > ___
> >
> > > **Q4.** Typo: DensNet @ Table 2
> >
> > **Ans:** Thank you for pointing out this typo. We will correct it in the revision.
> >
> > ___
> >
> > **Reference:**
> >
> > [1] Dan Hendrycks and Kevin Gimpel. A baseline for detecting misclassified and out-of-distribution examples in neural networks. ICLR, 2017.
> >
> > [2] Weitang Liu, Xiaoyun Wang, John Owens, and Yixuan Li. Energy-based out-of-distribution detection. NeurlPS, 2020.
> >
> > [3] Yiyou Sun, Yifei Ming, Xiaojin Zhu, and Yixuan Li. Out-of-distribution detection with deep nearest neighbors. ICML, 2022

---

> > > ### Author Response · Authors · 2022-11-26
> > > **Reply to Reviewer FQTf**
> > >
> > > Hello Reviewer FQTf, we would be grateful if you can confirm whether our response and revision have addressed your concerns and let us know if any issues remain. To recap our response,
> > >
> > > * **Ablation study on the ensembling strategy.** We have conducted ablation studies by comparing four ensemble schemes: BH, Naive, Average, and Voting. Please see Appendix D in the revised version.
> > >
> > > * **The comparison is not fair.** We use a model zoo to boost the performance of OOD detection and name our proposed method Zoo-based OOD Detection Enhancement (ZODE). To the best of our knowledge, there are no baseline methods focused on the ensembled OOD detector under the model zoo setting. Therefore, our experiments focus on validating the effectiveness of our ensemble method.
> > >
> > > * **Score function.** We consider three score functions: MSP, Energy, and KNN. The MSP score considers the softmax output. The energy score uses the logits. The KNN score is derived from the embedding distance. In the revision, we have added the detailed experimental results of ZODE-Mahalanobis.

---

### Official Review · Reviewer_Wjqr · 2022-10-24

**Confidence:** 4
**Correctness:** 3
**Technical Novelty And Significance:** 3
**Empirical Novelty And Significance:** 3
**Recommendation:** 6

**Clarity, Quality, Novelty And Reproducibility:**

The paper is clear and logically structured. Unfortunately, the code is unavailable, making it difficult to reproduce the experiments.
The idea of using an ensemble for OOD detection is simple but seems effective based on the paper's results. However, the idea is not entirely new, and the authors should compare the proposed model with other ensemble approaches. Therefore, the comparison with standalone methods seems not fair.
The analysis does not consider the computational cost added by the base classifiers.


**Details Of Ethics Concerns:**

I have no ethical concerns about the paper.


**Strength And Weaknesses:**

- The proposed approach is interesting and presents promising results.
- The idea of using an ensemble of classifiers for OOD is not new, but using the BH procedure to combine the base classifiers is clever. BH generally produces fewer Type I errors and performs best in sparse cases. What about the non-sparse cases?


- I miss some qualitative discussions. What is the impact of the pretrained models on the results? Is there a minimum quantity of base classifiers? Do they need to be trained in some specific dataset? Is it necessary that the training dataset of the base classifiers be related to the test samples? Does the increase in the number of base classifiers improve the results? Is there a limit to that? What is the impact of adding diversity in the base classifiers in this context?
- The authors do not consider the computational cost of using all base classifiers in the inference process when comparing the performance with the standalone SOTA methods. Could distilling the ensemble's knowledge using a distillation approach improve the process for computational cost purposes?
- Is the approach generic to another type of data or just work with images?
- How the SOTA methods were chosen? Are those the SOTA?
- The authors did not discuss the limitation of the proposed method. It will be meaningful to discuss the gap between the experiments in the current version of the paper and the real-world applications.


**Summary Of The Paper:**

This paper proposes the method ZODE, an ensemble scheme using pretrained models for OOD detection. The proposed scheme employs the Benjamini–Hochberg procedure to decrease the false discovery rate of OOD samples by combining the base model outputs. The authors conduct experiments on CIFAR10 and Imagenet datasets to evaluate the ZODE performance compared with SOTA ODD methods, and they show that the method improves the ODD detection performance.

**Summary Of The Review:**

The trustworthiness of ML approaches is significantly related to their ability to recognize out-of-distribution samples. This is an important topic with a lot of applications. The proposed paper presents one interesting approach but fails to analyze different perspectives of the problem. The authors focus on error and do not consider the computational cost of the proposed method or the comparison with other ensemble methods. The author should discuss the proposed approach's limitations and explore the qualitative aspects of the results.

---

> ### Author Response · Authors · 2022-11-13
> **Reply to Reviewer Wjqr (Part I)**
>
>
> We thank you for your time and valuable feedback. We are particularly grateful for your recommendations to improve the manuscript. We answer the questions here.
>
> * * *
>
> >  **Q1.** The idea of using an ensemble of classifiers for OOD is not new, but using the BH procedure to combine the base classifiers is clever. BH generally produces fewer Type I errors and performs best in sparse cases. What about the non-sparse cases?
>
> **Ans:** Thank you for the comments. In our experiments, there is an approximate non-sparse case: CIFAR10 vs LSUN (Table2). We present the results below. We can find that each single-model detector has a low FPR and high AUC. This implies that for each OOD input, multiple pre-trained models can identify it as an OOD sample. In this non-sparse case, our method performs well and still improves the best single-model results.
>
> |           | TPR   | FPR   | AUC   |
> | --------- | ----- | ----- | ----- |
> | ResNet18  | 95.00 | 18.50 | 96.84 |
> | ResNet18* | 95.00 | 1.78  | 99.48 |
> | ResNet34  | 95.00 | 10.22 | 98.39 |
> | ResNet50  | 95.00 | 7.10  | 98.83 |
> | ResNet101 | 95.00 | 6.65  | 98.90 |
> | ResNet152 | 95.00 | 7.22  | 98.88 |
> | DenseNet  | 95.00 | 7.90  | 98.60 |
> | Ensemble  | 94.96 | 1.50  | 99.61 |
>
> * * *
>
> > **Q2(1).** What is the impact of the pre-trained models on the results?
>
> **Ans:** We provide ablation studies to check the impact of the pre-trained models on the results (Table 2, Table 3b, and Table 5). We can find that the ensembled detector can significantly outperform all single-model detectors. This implies that no pre-trained model can dominate the performance of the ensembled detector. Therefore, in our results, there does not exist the case: the model zoo contains one high-quality pre-trained model which dominates the improvement of the ensemble.
>
> > **Q2(2).** Is there a minimum quantity of base classifiers?
>
> **Ans:** No. The number of pre-trained models can be very small. A special case is that the model zoo contains only one model. Then the proposed method becomes a single-model method, i.e., Algorithm 1 with $m=1$.
>
> > **Q2(3).** Do they need to be trained in some specific dataset?
>
> **Ans:** The answer depends on the detection score. If we use MSP or Energy score, the pre-trained model should be a classifier pre-trained on the ID data. For the KNN score, we only use feature extractors that can be pre-trained on different datasets.
>
> > **Q2(4).** Is it necessary that the training dataset of the base classifiers be related to the test samples?
>
> **Ans:** No. We assume that there is no auxiliary data and that the test samples are unseen.
>
> > **Q2(5).** Does the increase in the number of base classifiers improve the results?
>
> **Ans:** Intuitively, we can view the ensemble scheme as examining the test input from different perspectives. Here, one pre-trained model provides one perspective. Therefore, increasing the number of pre-trained models and adding diversity to the model zoo can improve the results. In general, high-quality pre-trained models with different structures are preferred. However, if the models are very similar, increasing the number of pre-trained models may only provide limited diversity and introduce more noise.  In this case, increasing the number of pre-trained models may degrade the results.
>
> > **Q2(6).** Is there a limit to that?
>
> **Ans:** Yes. There is a limit to the performance of the ensembled OOD detector. For example, if OOD data overlaps ID data, then the limit of FPR should be greater than 0. In the revision, we will present a theoretical example to study FPR in the appendix.
>
> > **Q2(7).** What is the impact of adding diversity in the base classifiers in this context?
>
> **Ans:** Please see the answer to **Q2(5)**.
>
> ***

---

> > ### Author Response · Authors · 2022-11-13
> > **Reply to Reviewer Wjqr (Part II)**
> >
> > ***
> >
> > **Q3.** The authors do not consider the computational cost of using all base classifiers in the inference process when comparing the performance with the standalone SOTA methods. Could distilling the ensemble's knowledge using a distillation approach improve the process for computational cost purposes?
> >
> > **Ans:** Thank you for your suggestion. We decompose ZODE into three stages: inference, test, and ensemble, and compare it with post hoc OOD detection in detail.
> >
> > **ZODE.**
> >
> > **Stage1.** Inference
> >
> > 1. Given a pre-trained model zoo $\{\phi_1,\ldots, \phi_m\}$, Compute the score value of $S(x_i, \phi_j)$, $\forall 1 \leq i \leq n$ and $\forall 1 \leq j \leq m$;
> >
> > **Stage2.** Test
> >
> > 2. Given a test input $x^*$, estimate the p-value of $x^{*}$ given $\phi_j$: Step 4 in Algorithm 1.
> >
> > **Stage3.** Ensemble
> >
> > 3. Sort $\{p_1, \ldots, p_m\}$ in ascending order: $\{p_{(1)}, \ldots, p_{(m)}\}$;
> >
> > 4. If $\exists 1\leq j \leq m$ such that $p_{(j)}\leq \frac{j}{m} (1-\text{TPR}_0) $, then $x^*$ is classified as an OOD sample.
> >
> > **Post Hoc OOD Detection.** (with one pre-trained model)
> >
> > **Stage1.** Inference
> >
> > 1. Given a pre-trained model $\phi$, compute the score value of $S(x_i,\phi), \forall 1\leq i\leq n$;
> >
> > 2. Compute the empirical distribution of $\{ S(x_i, \phi)\}_{i=1}^n$;
> >
> > 3. Determine a threshold $\lambda$ by the quantile of the empirical distribution.
> >
> > **Stage2.** Test
> >
> > 4. Given a test input $x^*$, compute the score value $S(x^*,\phi)$;
> >
> > 5. If $S(x^*,\phi)\leq \lambda$, then $x^*$ is classified as an OOD sample.
> >
> > We can find that
> > * The concern about the computational complexity comes from Stage 1, i.e. the inference procedure. For Stage 1, the computational complexity of ZODE is m times that of Post hoc OOD detection using a single pre-trained model, since ZODE uses m pre-trained models. In addition, Stage 1 only requires feedforward and is easy to parallelize. Therefore, the computational burden is not heavy.
> > * Stage 1 only needs the ID data and can be done before deploying the OOD detection algorithm. Therefore, the computational complexity of Stage 1 does not increase the test time when testing new inputs with ZODE.
> >
> > * * *
> >
> >
> > **Q4.** Is the approach generic to another type of data or just work with images?
> >
> > **Ans:** Thank you for the question. Using model zoos to improve single-model performance is a generic approach in both generalization and testing. In this work, we validate the utility of the model zoo for OOD detection and only consider image classification tasks. In future work, we will consider other tasks and more types of data.
> >
> > * * *
> >
> > **Q5.** How the SOTA methods were chosen? Are those the SOTA?
> >
> > **Ans:** In this work, we consider the OOD detection methods without auxiliary data. Our experiments focus on the baselines using single-model and single-detection methods. In this range, the KNN method is the SOTA.
> >
> > * * *
> >
> > **Q6.** The authors did not discuss the limitation of the proposed method. It will be meaningful to discuss the gap between the experiments in the current version of the paper and the real-world applications.
> >
> > **Ans:** Thank you for the comments. The main limitation of our method is that a lot of information needs to be passed from Stage 1 to Stage 2. The post hoc OOD detection only passes a threshold value to Stage 2, while our method requires passing the matrix of scores: $\{ S(x_i, \phi_j)\}$. Although this weakness does not affect the computational complexity of the test process, it does take up more storage space.
> >
> > * * *
> >
> > **Q7.** Unfortunately, the code is unavailable, making it difficult to reproduce the experiments.
> >
> > **Ans:**  We did not provide a public link to open our code in the original version. Please refer to the Supplementary Materials for the source code.
> >
> > * * *

---

> > > ### Author Response · Authors · 2022-11-26
> > > **Reply to Reviewer Wjqr**
> > >
> > > Hello Reviewer Wjqr, we hope that our response and revision have addressed the concerns raised in your review. If there remain concerns or if you have more questions, we are more than happy to provide additional clarification. To recap our response,
> > >
> > > * **Non-sparse cases.** In our experiments, there is a non-sparse case: CIFAR10 vs LSUN (Table 2). In this non-sparse case, our method performs well and still improves the best single-model results.
> > >
> > > * **Qualitative discussion.** In our response, we answer the questions point-by-point. In the revision, we have also added discussions about the FPR of the proposed method as the size of model zoo increases (m tends to infinity).
> > >
> > > * **Computational cost.** We have reorganized Algorithm 1 into three stages and illustrated that the computational burden is not heavy. Please see Section 3.3 in the revised version.
> > >
> > > * **Generic to another type.** Using model zoos to improve single-model performance is a generic approach in both generalization and testing. In this work, we validate the utility of the model zoo for OOD detection and only consider classification tasks. In future work, we will consider other tasks and more types of data.
> > >
> > > * **Limitation.** The main limitation of our method is that a lot of information needs to be passed from Stage 1 to Stage 2.

---

### Official Review · Reviewer_Ty7t · 2022-10-25

**Confidence:** 5
**Correctness:** 4
**Technical Novelty And Significance:** 2
**Empirical Novelty And Significance:** 2
**Recommendation:** 5

**Clarity, Quality, Novelty And Reproducibility:**

Please refer to the above section for clarity and novelty.
Some additional details are required to ensure reproducibility

**Strength And Weaknesses:**

Strengths:

1. Considering an ensemble of models to detect OODs is a simple yet effective idea.

2. The draft is clearly written and easy to follow.

3. Authors conducted extensive experiments of the standard benchmarks and the results are state of the art.

Weaknesses:

1. Overall novelty is limited as p-value based OOD detection is explored in conformal prediction literature.

2. What is the score function used in this work? This is important and needs to be discussed.

3. In OOD detection, FPR is more important than TPR. Does theorem 1 provide any guarantee on FPR?

4. Authors are encouraged to discuss the computational complexity of Algorithm 1. For each test sample, it requires computing the score
of all the training samples for all the models. This can be expensive.

5. Does the OOD detection performance depend on the quality of pre-trained models?

6. Some related references are missing

a. Cai, F.; and Koutsoukos, X. Real-time out-of-distribution detection in learning-enabled cyber-physical systems. ICCPS, 2020:
b. Kaur, R. et al., iDECODe: In-distribution equivariance for conformal out-of-distribution detection. AAAI, 2022

(a) considers p-value based conformal scores for OOD detection.
(b) proposed an approach to combine multiple p-values with a bound on TPR.

**Summary Of The Paper:**

This paper presents an approach to detect out-of-distribution (ODD) samples by an ensemble of pre-trained models. For a given test sample, OOD scores are computed based on a set of pre-trained models and a suitable p-value is determined whether the sample is OOD while maintaining a given true positive rate. experiments are conducted on common OOD benchmarks such as CIFAR-10, ImageNet, and Places365.

**Summary Of The Review:**

The idea of considering an ensemble of models for detecting OOD is interesting. However, the overall novelty is lacking. Please address the concerns in the weaknesses section.

---

> ### Author Response · Authors · 2022-11-13
> **Reply to Reviewer Ty7t (Part I)**
>
> We are thankful for your time and insightful comments. We are happy that you find our paper well-written and has solid experiments. We answer your questions here.
>
> ___
>
> >  **Q1.** Overall novelty is limited as p-value-based OOD detection is explored in conformal prediction literature.
>
> **Ans:** Thank you for the comments.
>
> - The novelty is positioned in the novel use of the model zoo and BH correction. To the best of our knowledge, we are the first to systematically study using a model zoo to boost OOD detection. Therefore, we name our proposed method **Z**oo-based **O**OD **D**etection **E**nhancement (**ZODE**). The p-value and BH correction here are tools used to exploit the model zoo.
> - Compared to the related OOD detection methods, our method simplifies the p-value estimators and multiple hypothesis test techniques. We directly compute the p-value according to its definition. Furthermore, we use a simple and fundamental technique (BH correction) of multiple hypothesis testing to ensemble multiple OOD detection decisions. We give theoretical guarantees and empirically verify that these simple tools work very well under the setup of the model zoo.
> - The proposed OOD detection method is interpretable. If a test input is classified as an OOD sample, our approach can point out which pre-trained models contribute to this decision. Please see Figure 1 and Figure 2 in Section 4.
>
> ____
>
> > **Q2.** What is the score function used in this work? This is important and needs to be discussed.
>
> **Ans:** In Section 4, we consider three score functions: MSP [1], Energy [2], and KNN [3]. The MSP score considers the softmax output.  The energy score uses the logits. The KNN score is derived from the embedding distance. On CIFAR10 benchmarks, we report the results of MSP vs ZODE-MSP, Energy vs ZODE-Energy, and KNN vs ZODE-KNN (Table 1). We find that ZODE-enhanced detectors consistently improve the performance of the corresponding baselines. Among the three enhanced detectors, ZODE-KNN outperforms the other two enhanced detectors and all baseline OOD detection methods. This observation is consistent with our motivation that leverages the diversity of pre-trained feature extractors. Therefore, on ImageNet benchmarks, we use the KNN score function.
>
> [1] Dan Hendrycks and Kevin Gimpel. A baseline for detecting misclassified and out-of-distribution examples in neural networks. ICLR, 2017.
>
> [2] Weitang Liu, Xiaoyun Wang, John Owens, and Yixuan Li. Energy-based out-of-distribution detection. NeurlPS, 2020.
>
> [3] Yiyou Sun, Yifei Ming, Xiaojin Zhu, and Yixuan Li. Out-of-distribution detection with deep nearest neighbors. ICML, 2022
>
>
> ___
>
> > **Q3.** In OOD detection, FPR is more important than TPR. Does theorem 1 provide any guarantee on FPR?
>
> **Ans:** Theorem 1 provides a guarantee of the reliability (TPR) of the proposed method and does not provide guarantees on the sensitivity to OOD samples (FPR). Following the existing works, we use experiments to validate the FPR of our proposed method. In general, it is not easy to theoretically discuss the FPR without additional constraints on OOD distributions, since the OOD distribution can be an arbitrary distribution different from the ID distribution.
>
> In Section 3.2, we prove that the p-values of an ID sample follow the uniform distribution $U[0,1]$.  If the test input is an OOD sample that can be detected by some pre-trained models in the model zoo, then the distribution of p-values shifts.
>
> Let $m$ be the number of all pre-trained models and $m_1$ be the number of the pre-trained models that can detect a test input as an OOD sample. Given an OOD input, there are $S$ pre-trained models (among the $m_1$ pre-trained models) that identify the input as an OOD sample. Then we study the average power $\mathbb{E}[\frac{S}{m_1}]$ since our method identifies a test input as an OOD sample if $S \geq 1$. We summarize the notations in the following table.
>
> |            | ID    | OOD   | Total |
> | ---------- | ----- | ----- | ----- |
> | Detect ID  | U     | T     |       |
> | Detect OOD | V     | S     | $R_m$ |
> | Total      | $m_0$ | $m_1$ | m     |
>
> We have
>  $$\mathbb{E} [\frac{S}{m_1}] = \mathbb{E} [ \frac{\frac{R_m}{m} (1-\frac{V}{R_m}) }{\frac{m_1}{m}}] \geq\frac{p_{*}(\alpha, F) (1-(1-\pi \alpha))}{\pi}$$
>
> as $m \rightarrow + \infty$, where $\pi=\frac{m_1}{m}$ and $\alpha = 1 - \text{TPR}_0.$
>
> According to [4], $p_{*}(\alpha, F)$ is the limit of the proportion of pre-trained models that identify the test input as an OOD sample. Therefore, if the average power $\mathbb{E}[\frac{S}{m_1}]$ is greater than $\frac{1}{m_1}$, then we can say that the power of our method is close to 1 with a high probability. In the revision, we will add more detailed discussions in the appendix.
>
> [4] Chi, Zhiyi. On the performance of FDR control: constraints and a partial solution. _The Annals of Statistics_ (2007).
>
> _____

---

> > ### Author Response · Authors · 2022-11-13
> > **Reply to Reviewer Ty7t (Part II)**
> >
> > _____
> >
> > > **Q4.** Authors are encouraged to discuss the computational complexity of Algorithm 1. For each test sample, it requires computing the score of all the training samples for all the models. This can be expensive.
> >
> > **Ans:** Thank you for your suggestion. The proposed method does not significantly increase the computational complexity.  Here we decompose ZODE into three stages: inference, test, and ensemble, and compare it with post hoc OOD detection in detail.
> >
> > **ZODE.**
> >
> > **Stage1.** Inference
> >
> > 1. Given a pre-trained model zoo $\{\phi_1,\ldots, \phi_m\}$,  Compute the score value of $S(x_i, \phi_j)$, $\forall 1 \leq i \leq n$ and $\forall 1 \leq j \leq m$;
> >
> > **Stage2.** Test
> >
> > 2. Given a test input $x^*$, estimate the p-value of $x^*$ given $\phi_j$: Step 4 of Algorithm 1.
> >
> > **Stage3.** Ensemble
> >
> > 3. Sort $\{p_1, \ldots, p_m\}$  in ascending order: $\{p_{(1)}, \ldots, p_{(m)}\}$;
> >
> > 4. If  $\exists 1\leq j \leq m$ such that $p_{(j)}\leq \frac{j}{m} (1-\text{TPR}_0) $,  then $x^*$ is classified as an OOD sample.
> >
> >
> > **Post Hoc OOD Detection.** (with one pre-trained model)
> >
> > **Stage1.** Inference
> >
> > 1. Given a pre-trained model $\phi$, compute the score value of $S(x_i,\phi), \forall 1\leq i\leq n$;
> >
> > 2. Compute the empirical distribution of $\{ S(x_i, \phi)\}_{i=1}^n$;
> >
> > 3. Determine a threshold $\lambda$ by the quantile of the empirical distribution.
> >
> > **Stage2.** Test
> >
> > 4. Given a test input $x^*$, compute the score value $S(x^*,\phi)$;
> >
> > 5. If $S(x^*,\phi)\leq \lambda$, then $x^*$ is classified as an OOD sample.
> >
> > We can find that:
> > - The concern about the computational complexity comes from Stage 1, i.e. the inference procedure. For Stage 1, the computational complexity of ZODE is m times that of Post hoc OOD detection using a single pre-trained model, since ZODE uses m pre-trained models. In addition, Stage 1 only requires feedforward and is easy to parallelize. Therefore, the computational burden is not heavy.
> > - However, Stage 1 only needs the ID data and can be done before deploying the OOD detection algorithm. Therefore, the computation complexity of Stage 1 does not significantly increase the computational cost when testing new inputs with ZODE.
> > ____
> >
> > > **Q5.** Does the OOD detection performance depend on the quality of pre-trained models?
> >
> > **Ans:** Yes. High-quality pre-trained models are preferred. In the CIFAR10 experiments, all pre-trained models have good ID prediction accuracy and perform well in OOD detection tasks. Then we can obtain great improvement in OOD detection by ensembling their OOD detection decisions. For ImageNet experiments, some pre-trained models performed poorly on ID prediction and OOD detection. Including these bad pre-trained models in the model zoo can weaken the performance of the ensemble. Therefore, we selected five top-ranked models in ID prediction and OOD detection to build the model zoo.
> >
> > ___
> >
> > > **Q6.** Some related references are missing:
> > >
> > > a. Cai, F.; and Koutsoukos, X. Real-time out-of-distribution detection in learning-enabled cyber-physical systems. ICCPS, 2020:
> > >
> > > b. Kaur, R. et al., iDECODe: In-distribution equivariance for conformal out-of-distribution detection. AAAI, 2022
> > >
> > > (a) considers p-value based conformal scores for OOD detection. (b) proposed an approach to combine multiple p-values with a bound on TPR.
> >
> > **Ans:** Thank you for the comments. We will add these two citations in the revised version.
> >
> > ___

---

> > > ### Author Response · Authors · 2022-11-26
> > > **Reply to Reviewer Ty7t**
> > >
> > > Hello Reviewer Ty7t, we would be grateful if you can confirm whether our response and revision have addressed your concerns and let us know if any issues remain. To recap our response,
> > >
> > > - **Novelty.** To the best of our knowledge, we are the first to systematically study using a model zoo to boost OOD detection. Our method simplifies the p-value estimators and multiple hypothesis test techniques. We give theoretical guarantees and empirically verify that these simple tools work very well under the setup of the model zoo. In addition, the proposed ensemble method is interpretable.
> > >
> > > - **Score function.** We consider three score functions: MSP, Energy, and KNN. The MSP score considers the softmax output. The energy score uses the logits. The KNN score is derived from the embedding distance.
> > >
> > > - **Theoretical guarantee on FPR.** We have added a theoretical guarantee on the power (i.e., 1-FPR) of the proposed OOD detection method. Please see Appendix B in the revised version.
> > >
> > > - **Computational complexity.** We have reorganized Algorithm 1 into three stages and illustrated that the computational burden is not heavy.

---

### Author Response · Authors · 2022-11-18
**General Response**

We thank the reviewers for their time and valuable and positive feedback.

We are encouraged that the reviewers found our method simple and effective (**R1**, **R2**),  interesting (**R1**, **R2**, **R3**), achieves SOTA results (**R1**), presents promising results (**R2**), and clearly outperforms other methods in most cases (**R3**). We are happy that they found our paper clear (**R1**, **R2**, **R3**),  easy to follow (**R1**),  logically structured (**R2**), and well-written (**R3**).

Two major concerns are computational complexity and lack of comparison with baseline ensemble schemes. For the computational complexity, we revised Algorithm 1 and focus on the inference stage to explain that the computation burden is not heavy. Please see our response and Section 3.3 in the revised version. For the ablation studies, we have conducted a few more experiments to answer the questions raised here. Please refer to Appendix D in the revised version.

Changes in revisions are marked in red. In the following, we list the main changes that we believe answer the reviewers' concerns and strengthen our contribution.

- We revised Algorithm 1 and added a paragraph discussing its computational complexity (Section 3.3).
- We added Appendix B to theoretically study the FPR of Algorithm 1 from an asymptotic perspective. The results account for the impact of pre-trained models and report the limit of FPR as the number of pre-trained models increases.
- In Appendix D, we added ablation studies on the choice of ensemble scheme and compared our method with three baseline ensemble schemes (Table 7, 8). Furthermore, we also emphasized the interpretability of our ensemble scheme (Section 3.3).
- In Section 3.3, we added a paragraph to discuss the limitations of Algorithm 1.
- In Table 1, we added the detailed results of ZODE-Mahalanobis.

We hope our response and revisions can address your concerns. We are happy to answer any further queries.

**Reviewer Acronym Reference:** We have used the order of the reviews as the reviewer's identity. Specifically, we have used the following: Ty7t as R1, Wjqr as R2, and FQTf as R3.

---

### Decision · Program_Chairs · 2023-01-20

**Decision:**

Reject

**Justification For Why Not Higher Score:**

All reviewers agreed after the discussion that the paper is borderline reject. The AC agrees with their concerns.

**Justification For Why Not Lower Score:**

n/a

**Metareview: Summary, Strengths And Weaknesses:**

**Summary**:
This paper proposes to ensemble multiple pre-trained models with different architecture by estimating and thresholding p-value for out-of-distribution detection. Experimental results show the effectiveness of the proposed method.
Overall, the paper has borderline mixed reviews (556).

**Strengths**:
- Considering an ensemble of models to detect OODs is a simple yet effective idea.
- The draft is clearly written and easy to follow.
- Theoretical analysis on the p-value looks interesting.

**Weaknesses**: The reviewers identified the following major weaknesses:

1. Limited novelty as p-values have been used in OOD detection before. [Ty7t]
2. missing related works (Cai & Koutsoukos, Kaur et al). [Ty7t]
3. Score function not described clearly. [Ty7t]
4. Since FPR is more important in OOD detection, does The 1 provide any guarantee on FPR? [Ty7t]
5. what is the computational complexity? [Ty7t, Wjqr]
6. Does the OOD detection performance depend on the quality of the pre-trained models? [Ty7t]
7. idea of using ensembles for OOD detection is not new, but using BH to combine base classifiers is clever. [Wjqr]
8. What about non-sparse cases? [Wjqr]
9. What is the impact of the pertained models on the results? Is there a minimum number of base classifiers? Does increasing the number improve results? What is the impact of diversity in the base classifiers? [Wjqr]
10. Could knowledge distillation on the ensemble be used to improve computational cost? [Wjqr]
11. Does it work with other types of data? or just images? [Wjqr]
12. How were the SOTA methods selected? [Wjqr]
13. What are the limitations (e.g., gap with real-world applications)? [Wjqr]
14. How does it compare to other ensemble methods? [Wjqr, FQTf]
15. No ablation study on the ensembling strategy (e.g., simple averaging or majority voting)
16. Why combine with MSP, Energy and KNN only? Does ranking of methods hold (ZODE-Mahal vs. ZODE-kNN)?



**Summary Of Ac-Reviewer Meeting:**

During the discussion, Reviewer FQTf raised the following issues:

- The comparison with other ensemble strategies is my major concern. As I know, model ensemble is a kind of default option for improving machine learning performances, so comparison with non-ensemble methods is not meaningful. So I expected to see a novel and/or effective ensemble strategy and comparison with baseline ensemble methods.
- Authors added Table 7 and 8 in Appendix D in the revision to address my concern, but it turned out that the performance of the proposed method is not so better than baseline ensemble methods, in terms of AUC.
- Ensembling based on the p-value and corresponding analysis are interesting, as Reviewer Wjqr said, but by looking at the performance, I can't see the reason why we need such a sophisticated ensemble strategy.
- authors also claim that TPR is "out of control" and show FPR under somewhat random TPR for other ensemble methods, which I do not fully understand what they actually mean by; maybe they observed discrete changes on TPR?
- Also, I think authors count the fact that "their work is first to take advantage of (publicly available) model zoo for OOD detection" as the main contribution, but it seems no reviewers agreed on this.

Reviewer Wjqr thought that the paper had potential, but agreed with the weaknesses raised by the other reviewers, making the paper into borderline to reject.